# Evaluating Alternatives to Water as Solvents for Life: The Example of Sulfuric Acid

**DOI:** 10.3390/life11050400

**Published:** 2021-04-27

**Authors:** William Bains, Janusz Jurand Petkowski, Zhuchang Zhan, Sara Seager

**Affiliations:** 1Department of Earth, Atmospheric, and Planetary Sciences, Massachusetts Institute of Technology, Cambridge, MA 02139, USA; jjpetkow@mit.edu (J.J.P.); zzhan@mit.edu (Z.Z.); seager@mit.edu (S.S.); 2School of Physics & Astronomy, Cardiff University, 4 The Parade, Cardiff CF24 3AA, UK; 3Department of Physics, Massachusetts Institute of Technology, Cambridge, MA 02139, USA; 4Department of Aeronautics and Astronautics, Massachusetts Institute of Technology, Cambridge, MA 02139, USA

**Keywords:** alternative biochemistry, alternative solvents, sulfuric acid biochemistry, sulfuric acid reactivity, sulfonation, solvolysis

## Abstract

The chemistry of life requires a solvent, which for life on Earth is water. Several alternative solvents have been suggested, but there is little quantitative analysis of their suitability as solvents for life. To support a novel (non-terrestrial) biochemistry, a solvent must be able to form a stable solution of a diverse set of small molecules and polymers, but must not dissolve all molecules. Here, we analyze the potential of concentrated sulfuric acid (CSA) as a solvent for biochemistry. As CSA is a highly effective solvent but a reactive substance, we focused our analysis on the stability of chemicals in sulfuric acid, using a model built from a database of kinetics of reaction of molecules with CSA. We consider the sulfuric acid clouds of Venus as a test case for this approach. The large majority of terrestrial biochemicals have half-lives of less than a second at any altitude in Venus’s clouds, but three sets of human-synthesized chemicals are more stable, with average half-lives of days to weeks at the conditions around 60 km altitude on Venus. We show that sufficient chemical structural and functional diversity may be available among those stable chemicals for life that uses concentrated sulfuric acid as a solvent to be plausible. However, analysis of meteoritic chemicals and possible abiotic synthetic paths suggests that postulated paths to the origin of life on Earth are unlikely to operate in CSA. We conclude that, contrary to expectation, sulfuric acid is an interesting candidate solvent for life, but further work is needed to identify a plausible route for life to originate in it.

## 1. Introduction

The chemistry of life requires a dense fluid phase in which the molecules of life can react [1]. Speculation on the biochemistry on other worlds usually assumes that the solvent for life is water [2,3,4]. However, speculation about other solvents persists, including the possibility of life in liquid methane and nitrogen, ammonia, and ammonia-water mixtures, and in supercritical fluids [5,6,7,8,9,10].

The recent tentative detection of phosphine in the atmosphere of Venus [11] has revived interest in the possibility of life in the clouds of Venus. If life exists on Venus, it is likely that it inhabits the clouds that permanently cover the planet, as the surface is inhospitable [12,13,14,15,16,17,18]. The clouds are believed to consist of photochemically-generated sulfuric acid, varying from ~70% to > 100% sulfuric acid [19,20] (discussed in Section 4.1 below). Ballesteros et al. [21] showed that planets hosting substantial surface sulfuric acid could be as abundant as planets hosting surface liquid water. Thus, both solar system relevance and potential cosmic abundance lead us to consider whether sulfuric acid could fulfill the many criteria required for it to act as a solvent for life.

The problems facing life living in concentrated sulfuric acid have generally been ignored in speculations about life on Venus, with implications that such an environment is analogous to acid environments on Earth (as in, for example, “at the very least, then, the cloud decks of Venus offer an aqueous environment for colonization by life.” [22]). Such comparisons are not accurate. Concentrated sulfuric acid is different in kind from aqueous solutions of sulfuric acid, as discussed below. Life in concentrated sulfuric acid could exist by maintaining an aqueous internal phase in which an Earth-like biochemistry could exist. This is the model assumed by Bains et al. [23], and implicitly assumed in studies that draw parallels between terrestrial acidophiles and the Venusian environment. How a barrier between an aqueous cell interior and an environment of concentrated sulfuric acid could be established and maintained is unknown and without precedent in any terrestrial extremophile. Also unknown is how life could originate in sulfuric acid.

Alternatively, life could use concentrated sulfuric acid as a solvent, and as a necessary consequence use different chemistry from terrestrial life. This paper is the first step in exploring the possibility of utilizing concentrated sulfuric acid as a solvent for life. We emphasize that throughout the paper we are discussing concentrated sulfuric acid (CSA), i.e., a liquid that is H_2_SO_4_ with other molecules dissolved in it as minor species. We evaluate the potential of sulfuric acid to act as a solvent for life in terms of its ability to stably dissolve some, but not all, molecules from a structurally and functionally diverse chemical space, and similarly to stably dissolve some, but not all, polymers (further details of the reason for these criteria are provided in Appendix A). We review the literature on solubility, stability, and other properties of sulfuric acid, and show that the potential lack of stable diversity of chemicals in sulfuric acid is the principal reason for doubting that it could be a solvent for life (Section 2). We then describe methods (Section 3) used to provide a quantitative analysis of the stability of substances in CSA using the clouds of Venus as a case study (Section 4). We conclude that complex chemistry capable of supporting a biochemistry is not ruled out for CSA as a solvent, but that an origin of such life is difficult to model (Section 5).

## 2. Background on CSA as a Potential Solvent for Life

### 2.1. Chemical Reactivity of Molecules in CSA

Sulfuric acid is a well-known laboratory and industrial reagent. It is a strong acid, a powerful dehydrating agent and a strong oxidizing agent. Its ability to reduce dry sugar to steaming char is documented in many science demonstration videos. A chemist’s immediate response to the concept of sulfuric acid as a solvent for life is likely to be that sulfuric acid is too reactive, and that insufficient chemical function or diversity could be stable in the presence of sulfuric acid.

The chemistry of CSA is the chemistry of protonation, dehydration, and attack by SO_3_ (sulfonation). CSA’s acidity (Hammett acidity ~ −11.5) results in many compounds forming protonated, positive ions on solution. These include alcohols, aldehydes, ketones, alkenes, phenyls, and naphthalenes [24], as well as more familiar bases such as amines. Some of the resulting ions are stable, and can be recovered in their original, non-conjugate form by diluting the acid. Other compounds form protonated ions which subsequently react. Alcohols are often sulfated—whether they are then dehydrated to ethers depends on the alcohol and the temperature. Thus, Rosenbaum and Symons comment that it is well-known that “solutions of many alcohols and olefins in sulfuric acid become yellow on standing” due to the activation and cross-linking of the molecules [25]. We note that aldehydes in solution in CSA are in equilibrium with the enol tautomer, and are not almost exclusively in the keto form as it is in water. Some aldehydes can be stably dissolved in CSA, many cannot.

A range of compounds react in CSA to give poorly defined products. For example, 2,4-hexadienal turns brown and then black almost immediately on addition to >50% H_2_SO_4_ at RT [26], as does isoprene [27]. Glyoxal is also highly reactive, forming complex products [28], and 1,2-diols are rapidly dehydrated, as are some beta-hydroxy acids upon heating in CSA [29]. Thiols are readily oxidized in CSA at 298 K (referred to in [30]), although the kinetics of the oxidation of thiols in concentrated sulfuric acid is not well-studied. Heptanal and octanal can form stable solutions in CSA at a low concentration and at low temperature; in concentrated aldehyde solution at elevated temperature, both react to form multiple products [26,31,32]. We note, however, that the complexity of the products of the reaction with CSA does not concern us in this paper, as we are concerned here with the rate of disappearance of the starting material, not the rate of formation of a specific product.

By contrast, a range of chemicals react reversibly with sulfuric acid. We consider chemicals undergoing these reversible reactions as stable to reaction with CSA. For example, alcohols dissolve and then react slowly at room temperature to form sulfates [33,34]. However, the basic topology of the molecule is unchanged by this reaction. We consider these readily reversible reactions to be equivalent to the protonation of amines and aromatic compounds alluded to above, and functionally equivalent to the ionization of acids or the fractional hydration of aldehydes in water.

Thus, there is a range of stability of chemicals in CSA, from extremely stable to extremely unstable. Exploration of the potential of CSA as a solvent for life therefore requires quantitative enumeration of how many member molecules of different chemical spaces (represented here by different repositories of organic molecules) are stable and unstable in CSA, for different stability criteria. We perform this quantitative analysis in Section 4.

### 2.2. Solubility of Molecules in CSA

Concentrated sulfuric acid is notorious for its ability to dissolve organic material. While a wide range of compounds react rapidly with hot CSA (Figure 3), at low temperatures these reactions are much slower or altogether absent. While pure sulfuric acid freezes at 283.6 K, aerosols of CSA can remain liquid down to 170 K for 40% H_2_SO_4_, and 240 K for 60%–90% H_2_SO_4_, [35,36]. Thus, there is a possibility that a range of compounds are stably soluble in cold sulfuric acid solution in the droplets in the colder upper clouds of Venus.

In general, nitrogen-containing compounds such as amines, pyridines, and azoles are stably soluble in CSA, as they readily protonate to form stable cations, which are then resistant to electrophilic attack by SO_3_. By contrast, alcohols are protonated to oxonium derivatives, but these are relatively unstable and prone to rearrangement such as dehydration. Thus, a wide range of amines stably dissolve in CSA as ammonium salts [37,38,39], and pyridines and pyrimidines are only sulfonated by oleum or CSA at temperatures above 150 °C [40,41,42]. Polyaromatic hydrocarbons and their quinones in general form stable solutions in CSA as colored, protonated species [43,44], unlike solution in octane (which solutions are uncolored) or water (where they do not dissolve). Some aromatic hydrocarbons dissolve in CSA to give paramagnetic species (i.e., positive ions), through nucleophilic attack of ring electrons on the sulfur atoms in SO_3_ [45]. In some cases, nucleophilic attack is completely reversible when the solution is diluted, while in other cases, it results in irreversible sulfonation of the aromatic ring. Figure 1 provides an overview of the examples of the diverse solubility behavior of organic chemicals in CSA.

However, as with any chemistry, specifics of the molecular structure can modulate reactivity and solubility behavior within general trends. For example, among similar structures of polyaromatic ring systems (Figure 1), pyrene is very soluble and decomposes in CSA [46]; anthracene [47], perylene, tetracene, and 3,4-benzopyrene [46] give stable colored and paramagnetic solutions; while naphthalene, phenanthrene, and chrysene are insoluble [45] (note that the papers reporting these results do not give the quantitative solubility values required for solubility prediction). Carboxylic acid chlorides form stable solutions in CSA [24] (unlike in water, where they are rapidly hydrolyzed). Some phosphines form stable solutions in CSA. Perfluoro aromatic hydrocarbons are insoluble in CSA as they are in water, but heptafluoroquinoline is soluble, dissolving as a protonated form [48].

Alkanes are relatively insoluble in CSA, as illustrated by the early 20th century method for measuring the fraction of alkenes in petroleum distillates by shaking it with CSA: alkenes dissolved almost quantitatively, whereas alkanes did not [49]. Triglycerides are also insoluble even in hot (>80 °C) concentrated sulfuric acid [50].

Thus, there is substantial evidence that molecules possessing a range of solubility properties in CSA exist, from highly soluble amines to insoluble naphthalene. Such diverse solubility behavior of many different molecules is one of the critical characteristics needed for any biochemistry.

#### Amphipaths in CSA

The existence of substances that are soluble in CSA and ones that are insoluble suggests that molecules combining both characteristics, i.e., partially soluble CSA amphiphiles, should exist. There is some initial evidence that molecules that are amphipathic to CSA do exist. One study showed that butanol and hexanol concentrate on the surface of 70% H_2_SO_4_, and substantially reduce its surface tension, especially at 204 K [31], a property consistent with them acting as amphipathic surfactants at these low temperatures. Nonanal is similarly poorly soluble in CSA, but taken up rapidly and efficiently into CSA mixed with hydrophobic oleic acid, also suggesting the potential for the existence of stable hydrophobic phases in CSA [51]. Whether genuine micelles or ‘membranes’ can be formed in sulfuric acid is a question for future dedicated experimentation and modelling.

We therefore conclude that a range of chemicals can form stable solutions, as free molecules or as conjugate acids, in CSA, that some are insoluble, and there is suggestive evidence that CSA amphipaths can exist.

### 2.3. Polymer Stability and Solubility in CSA

There is a rich literature on the stability and solubility of polymers in CSA, both because of the use of sulfuric acid in synthesis and modification of polymers, and because of its use as a solvent for spinning and spin-coating polymers that are insoluble in water or organic solvents.

Appendix B summarizes the stability and solubility of a selection of polymers in CSA. A number of the polymers listed are sulfonated on aromatic rings at elevated temperatures (e.g., [52,53]); however, this reaction is extremely slow below 273 K. It is notable that despite the fact that proteins are rapidly denatured and specific functional groups in amino acid side chains are attacked in concentrated sulfuric acid [54] the peptide bonds themselves are fairly stable in CSA. For example, polyglycine [54,55] and nylons [56,57] are sufficiently stable in 96% sulfuric acid for it to be used as a solvent in NMR studies, and the aromatic polyamide Kevlar is highly soluble and stable for months at room temperature in CSA [58].

To an extent, both solubility and stability depend on the physical form of the polymer, which itself depends on its route of synthesis. Thus, for example, the polyanthraquinones described by [59] are classified as soluble or insoluble in CSA depending on the solvent used to synthesize them. The stability of insoluble polymers depends almost entirely on their physical form; for example, cross-linked polystyrene is insoluble and effectively stable in CSA, but the same polymer swollen with dichloroethane is sulfonated, but not hydrolyzed, in CSA at 100 °C [60].

We conclude that there are a range of polymers that are soluble in CSA, and others that are not soluble, meeting the requirements for the diverse solubility behavior of polymers in any biochemistry.

### 2.4. Other Properties of CSA

We briefly review the other properties of CSA that might be relevant to its potential as a solvent for life.

**Viscosity.** CSA is more viscous than water, showing a steep rise of viscosity with concentration up to a maximum of 2.5 × 10^−2^ Ns/m^2^ at 85% sulfuric acid at 25 °C (by contrast, water is 8.91 × 10^−4^, i.e., ~30 times less viscous). The viscosity increases sharply with decreasing temperature [61], such that the extrapolated viscosity of pure sulfuric acid at 240 K is 0.225 Ns/cm^2^. Diffusion-limited reactions may therefore be substantially slower in cold sulfuric acid.

**Dielectric constant.** Sulfuric acid is a strong dielectric, with a dielectric constant of 110 (vs water 84). Dielectric constant is important for solvating charged species, and for enabling some of the macromolecular interactions that stabilize proteins and nucleic acids in terrestrial biochemistry [1]. In this regard, CSA does not differ substantially from water (obviously, in other regards the two are very different, as discussed extensively above).

**Solvation of metals.** There is an extensive literature dating back over 100 years about the study of metal salts in CSA (reviewed in part in [24]). In regards to CSA’s ability to dissolve metals, CSA is superior to water. Many metal ions such as Fe(II) may be oxidized in CSA, but complexes of Fe(II) such as Fe (II) phthalocyanin can stably dissolve in CSA [62], providing the potential for Fe(II)/Fe(III) redox chemistry [63].

### 2.5. Conclusion of the Literature Survey on the Properties of CSA

From our review of the known chemical properties of CSA, we conclude that the principal limitation on CSA as a potential solvent for the complex biochemistry of life is likely to be the stability of diverse chemicals in CSA. We therefore explored the stability of diverse chemicals in CSA, as described in the next two sections.

## 3. Materials and Methods

We present the datasets and models that we used to model the stability of compounds in CSA. This falls into two parts. The first is the method of prediction of stability of a chemical, which is based on literature reports of kinetics in sulfuric acid (Section 3.1). This includes defining conditions in Venus’s clouds (Section 3.2). The second part is the definition of databases representative of chemical space, and the software used to handle those databases (Section 3.3).

### 3.1. Reactivity Prediction Methods

There is a rich literature on the kinetics and mechanism of reactions in sulfuric acid [24,64,65]. An accurate prediction of the rate of reaction requires knowledge of the basicity of the reacting compounds, and the mechanism of reaction (see for example [66,67,68,69]). However, both basicity (and hence protonation state in CSA) and mechanism are usually determined from the kinetics of reaction. For new compounds, the reaction mechanism and basicity cannot be accurately predicted, and hence kinetics cannot be predicted without the mechanistic insight that is usually derived from kinetic measurements. We therefore took an empirical approach of analyzing the observed rate of reaction of a diverse set of molecules in a variety of sulfuric acid concentrations and then deriving general rules about the reactions of *types* of molecules from this collection of specific rates. We recognize that this is an approximation, but believe that it is suitable for the order-of-magnitude predictions that are useful.

Data were collected from 286 literature studies (listed in [70]) of the reaction kinetics of organic molecules with concentrated sulfuric acid. These were used to estimate the reaction rate constants as a function of acid concentration and temperature for the functional groups in those molecules.

This data set and its creation and applications has been described in detail in [70]. In [70], we provide a detailed description of the methods used to abstract kinetic data from the literature, including a complete list of chemicals, reactions, and literature sources. To summarize [70], reaction rates were converted to uniform units. The reactive group in the molecule was classified as one of a set of 130 ‘Functional Groups’, or as a sulfonation target. The change in reaction rate with acid concentration and with temperature was then drawn from the original data presented in each literature study. Values that were not provided in a specific study were interpolated from other reaction studies of that Functional Group to provide a full matrix of reaction rate constants for each Functional Group. For the purposes of this study, the rate constants of reactions proceeding in various sulfuric acid concentrations (derived as described below) and temperature for the atmospheric profile of Venus were used to calculate the predicted rate constants. See [70] for further details of this method.

It is widely found that both solvolysis and sulfonation of substance in CSA is a pseudo-first order process, that is the rate of reaction is given by Equation (1):(1)$$\frac{\partial \left[X\right]}{\partial t}=-k\cdot \left[X\right]$$
where [*X*] is the concentration of the substance under attack, *t* is time, and *k* is the rate constant. Reaction is only *pseudo*-first order because change in sulfuric acid concentration will substantially change the rate. However, for a constant sulfuric acid concentration, as would occur when there is a large excess of acid over other reagents, Equation (1) above holds.

The stability of a molecule in CSA was then determined as follows. The occurrence of Functional Groups within molecules was determined using a custom python code, based on the RDKit Cheminformatics sub-structure matching module, as described previously [71,72]. The rate for each Functional Group in a molecule was determined as above, and the half-life of that reaction was calculated using Equation (2):(2)$$t=-\frac{ln\left(0.5\right)}{k}$$
where *t* is the half-time in seconds and *k* is the reaction rate constant in seconds^−1^. The fastest of the rates (i.e., the shortest half-time) for all the Functional Groups in a molecule was used as an estimate for the rate at which the entire molecule was broken down or sulfonated in sulfuric acid. In other words, if a molecule contained two Functional Groups, it was assumed that the molecule broke down at the rate of reaction of the fastest reacting of those groups. For the case when one Functional Group is present in several places in a molecule (e.g., alcohol groups in sugars), the half-life of the molecule overall was corrected according to Equation (3):(3)$$t=-\frac{ln\left(1-\frac{0.5}{N}\right)}{k}$$
where *t* is the half-time of reaction, *k* is the rate constant, and *N* is the number of copies of a reactive group in the molecule. This assumes that the reaction of the groups was independent; where this is manifestly not, so (e.g., for 1,2-diols) a separate Functional Group that comprises two smaller Functional Groups was created.

As noted above, the model used here can predict reaction rates to an order of magnitude only. However, the exact prediction of kinetic parameters is not a goal of this study. Our purpose in doing such an extrapolation is two-fold. First, our goal is to provide a first pass estimate of the chemical reactivity (and hence stability) of molecules in concentrated sulfuric acid. Second, our goal is to demonstrate that such an analysis, even if approximate, is both useful and informative, and hence encourage further laboratory studies of the kinetics of these and other reactions in sulfuric acid so as to approximate possible exoplanet conditions more accurately.

Our model does not take account of pressure. Pressure changes the rate of a reaction if there are volume changes or solvent order changes around the transition state(s) of a reaction [73]. The effect of pressure on reaction rates in CSA have not been systematically explored, but for organic reactions in water, the relevant volume changes imply that the change from atmospheric pressure to the critical pressure of water (217 atmospheres) will at most change the rate of a hydrolytic reaction by a factor of 20%, and change from the sea’s surface to the pressure at the bottom of the Challenger Deep (10,900 m, pressure of 1100 atmospheres) would speed up hydrolysis ~2.3 times [73,74,75,76]. Pressure will affect the *concentration* of a volatile substance in CSA, but not the first order reaction kinetics of that substance.

### 3.2. Concentration of Sulfuric Acid in the Clouds of Venus

As the concentration of water in Venus’s atmosphere is known [77,78], but the concentration of sulfuric acid in the cloud droplets is less constrained [19,78,79,80], we inferred the concentration of sulfuric acid by assuming that water vapor in the clouds was in equilibrium with water in the sulfuric acid. The relationship between the water vapor pressure over sulfuric acid and acid concentration and temperature was taken from [81], from which a general equation was derived empirically (Equation (4)):(4)$$P=\frac{ln\left(ppH_{2}O\right)-15.988-\frac{42.2}{T}}{0.053346-\frac{84.03}{T}}$$
where *P* is the percentage of sulfuric acid, *ppH_2_O* is the partial pressure of water in millimeters of mercury, and *T* is the absolute temperature. For each altitude, a concentration of sulfuric acid was calculated which was consistent with the observed partial pressure of water at the observed temperature.

### 3.3. Chemical Datasets and Software

We used several sets of molecular structures in this study. We summarize their source and processing here.

We wished to explore the extent to which the space of complex chemistry was stable in CSA. To do this, we needed sets of chemicals that represent all of this ‘Chemical Space’. We hypothesized that chemicals that were stable in water might have different stability properties in CSA to chemicals that were not stable in water, and we therefore chose sets of chemicals that were likely to include members that had different stability profiles towards hydrolysis by water.

We chose three sets of chemicals that represented the full breadth of human chemical ingenuity as a representation of the space of chemicals that are stable at room temperature and pressure (although not necessarily stable to the terrestrial oxygen-rich atmosphere). We compiled a collection of molecules that are likely to be stable to hydrolysis in water as follows. Molecules were extracted from the ChEMBL database (https://www.ebi.ac.uk/chembl/ (accessed on 30 July 2020)) [82]. ChEMBL is a data resource of chemicals linked to biological data, and so is strongly biased towards molecules that have been considered as drug candidates. All molecules with screening data for activity against cholinesterase, peptidases, proteases, or beta-lactamases were extracted, and duplicates removed. If screening data was available for a molecule, this implies that molecule had been screened in a biological assay, and hence was stable in water for a timescale of at least a few minutes. (Any set of assay results could be used for this purpose; the result we require is not the result of the assay, but the fact that a biological assay has been performed, which implies sufficient stability in water.)

We collected a wider space of chemicals, many of which are expected to be reactive towards water, from a set of the SigmaAldrich catalogue (kindly supplied by Bret Daniel (SigmaAldrich)). SigmaAldrich (https://www.sigmaaldrich.com/ (accessed on 5 September 2020)) is a leading aggregator of scientific supplies, and as such provides a very wide range of reagents for research.

Our third dataset was the PAN Pesticide databases (https://pesticideinfo.org/ (accessed on 29 July 2020)) [83]. PAN compiles pesticides (which are stable to hydrolysis in water) and industrial chemicals used in their synthesis or formulation (which may not be water stable or soluble). As such, PAN provides an intermediate in likely water solubility and stability between ChEMBL and SigmaAldrich data sets of molecules.

We obtained the list of water-soluble molecules present in the Murchison meteorite from [84].

All these four sets of molecules are provided coded in SMILES [85] strings in Appendix A.

Lastly, we used a set of approximately 220,000 molecules produced by living organisms (‘natural products’), which has previously been described in [86]. As this database has been compiled from sources including commercial databases available to MIT under limited license, it cannot be made publicly available; however, the authors welcome opportunities to collaborate on its use. See Table 1 for an overview of used chemical databases.

Combimol-B has been described in [87], and is available for non-commercial use from WB. This program generates a systematic set of all chemicals that can be built from a defined set of atoms, bonds, and molecular topologies. For this study, Combimol-B was run with all possible topologies from 3–8 atoms, but only including rings of 5 or more atoms, and with the atoms specified in the figure legend of Figure 8. The molecules generated by Combimol-B for this study are also included in Appendix A.

RDKit is a general-purpose computational chemistry toolkit in Python. RDKit is described in [88], and is available from http://www.rdkit.org (accessed on 18 July 2020). AllChemy is described in [89] and is available at https://life.allchemy.net/ (accessed on 18 July 2020). Chemical database handling and visualization was performed using PerkinElmer ChemOffice suite under site license to MIT.

## 4. Results

We present the results of our analysis of the stability of chemicals in CSA. As exemplars of diverse chemistry, we use the three chemical databases described in Section 3.3., as well as a set of Natural Products representing the chemistry of terrestrial life (Table 1). We use the clouds of Venus as a case study of an environment where CSA is present as a liquid and where life has been speculated to also be present. However, our approach could be readily applied to other environments apart from Venus’s clouds.

### 4.1. Concentration of Sulfuric Acid in the Clouds of Venus

Venus’s clouds are inferred to be composed of sulfuric acid droplets. The presence of sulfuric acid is supported by:(a)photochemical models of the atmosphere, which predict SO_3_ and H_2_SO_4_ to be present throughout the atmosphere [77,78];(b)measured and modelled levels of H_2_O, which together with SO_3_ would efficiently form H_2_SO_4_ [77,90,91];(c)measured levels of gaseous H_2_SO_4_ measured by microwave spectrometry [20];(d)inferred refractive index of the cloud droplets [79,80], which is consistent with the clouds being made of at least 70% sulfuric acid in water.

At cloud temperatures, H_2_SO_4_ would be expected to condense into liquid. However, it is unlikely that the clouds are pure sulfuric acid. Dissolved sulfur species and CO_2_ could change the refractive index, and if there are phosphorus species present in the atmosphere, then these would also be present in solution in the droplets. We therefore estimated the concentration of the sulfuric acid in cloud droplets by assuming that the droplets were in equilibrium with the water in the atmosphere around them. Atmospheric water abundance as a function of altitude was taken from [78] and the atmospheric temperature profile from Venus International Reference Atmosphere (VIRA).

The concentration profile is shown in Figure 2. The clouds usually are considered to extend from ~47 km to 70 km [19]; we therefore carried out calculations from 45 km to 70 km. Concentrations of >100% imply ‘fuming’ sulfuric acid or oleum (H_2_S_2_O_7_, or a solution of SO_3_ in H_2_SO_4_; see [24] for a summary of the species in sulfuric acid as a function of concentration). This profile was used for all subsequent calculations of chemical stability.

### 4.2. Quantitative Modelling of Stability of Chemical Compounds in CSA

Any calculation of properties of the molecules in a ‘chemical space’ must have example molecules on which to work. In this work, we are concerned not just with chemicals made by terrestrial biochemistry but also with chemicals that could be components of an unknown biochemistry that functions in CSA. Obviously, those components are unknown, and so cannot be used as examples of chemical space. We therefore chose three diverse sets of chemicals derived from human chemistry to probe the space of possible stable chemistry, as described in Table 1 and in the Methods section (Section 3.3). These are test examples of a diverse chemical space of mostly organic chemicals.

We modelled the stability of the four sets of molecules (Table 1) described in the Methods Section above (Section 3.3) for their stability in sulfuric acid. A summary of the results is shown in Figure 3. The results suggest that biological molecules (Natural Products; NP dataset on Figure 3) are quite unstable in sulfuric acid, but human-synthesized chemicals are much less unstable (Sigma, PAN, ChEMBL on Figure 3), especially at higher altitudes in Venus’s atmosphere (i.e., lower % acid and lower temperatures).

The curves in Figure 3 shows a steep rise in half-life, with rising altitude up to the mid-cloud level (~60 km), but then a pronounced dip at ~70 km (roughly the cloud top). A cause for this steep rise in half-life of chemical can be suggested based on the values presented on Figure 2. Up to ~60 km, both temperature and concentration of CSA decrease steeply with increasing altitude, resulting in a steep rise in half-life of chemicals with increasing altitude, as reaction rates generally (but not always) increase with increasing concentration of sulfuric acid, and always increase with increasing temperature. Above 60 km, the decline in temperature with altitude is less pronounced and the changes in sulfuric acid concentration dominate changes in the reaction rate; a slight peak in sulfuric acid concentration at ~70 km corresponds to a dip in stability. The slight rise in sulfuric acid concentration at ~70 km is itself the effect of the very steep decline in water vapor concentration at that altitude compared to temperature.

The average of half-lives of chemicals presented in Figure 3 hides substantial variability. Figure 4 summarizes the number of compounds in the Natural Products and CheMBL databases by predicted half-life in CSA as a function of altitude in Venus’s atmosphere.

While the large majority of Natural Products are unstable (in that they have predicted half-lives of less than a second at any altitude), a small fraction of Natural Products are stable above 55 km. These compounds are predominantly amines and polyamines (e.g., methylamine, spermine, spermidine), and compounds with inherent positive charge (e.g., taurine, choline, dimethyl-s-propionate) which dissolve stably in CSA [37,38,39]. Similarly, while most of the ChEMBL database is predicted to be stable above 60 km, some have a half-life of <1 s even at 70 km.

The fraction that are ‘stable’ depends on the definition of stability. Some metabolites have half-lives of a few seconds in normal metabolism, while the components of some proteins are required to last 70 years in humans without turnover or repair (e.g., [92,93]). Whether chemicals are stable enough to be potential components of a biochemistry therefore depends on their role in biochemistry. We therefore analyzed the fraction of each dataset that is stable by different criteria, averaged across altitudes above 62 km. The results are shown in Figure 5.

Figure 5 shows that between 25% and 35% of the non-biochemical sets of chemicals are predicted to have a half-life of at least 10^8^ s (~3 years). Thus, in principle, long-term stability of a diverse range of molecules playing structural or genetic roles in concentrated sulfuric acid is not impossible.

The instability of terrestrial biochemistry to concentrated sulfuric acid is not a surprise. The relative stability of synthetic chemicals in CSA is more surprising. Anecdotally, we have found that organic chemists eschew the use of sulfuric acid as a reagent because of its tendency to form tar as a product of reactions; as one report put it, “unfortunately, we were unable to convert [products] into acids, for under the action of concentrated sulfuric acid at 100 °C quantitative carbonization occurred” [94].

It therefore might be suggested that the only chemicals surviving in concentrated sulfuric acid are simple or monotonous–e.g., polyaromatic hydrocarbons or alkanes. As discussed in the Appendix A, such chemical monotony is not compatible with the chemistry being the basis of life (see Appendix A for details). We therefore explored the degree to which stable compounds in the non-biochemical datasets represented a diverse set of chemicals.

### 4.3. Diversity of Stable Chemistry in Sulfuric Acid

We took two approaches to characterizing chemical diversity in concentrated sulfuric acid, and specifically in the environment of Venus; the first is counting chemical bonds, and the second is counting elemental abundance in stable or unstable molecules.

The number of different types of chemical bonds in a set of chemicals is a rough indication of the diversity of structures in that set of molecules. Thus, alkanes have only two types of bonds (C-H and C-C), alkenes three (C-H, C-C and C=C), amides five (C-H, C-C, C=O, C-N and N-H), and so on. For each of the four datasets, we computed the average number of types of bonds in molecules that are stable at each altitude in the Venus model atmosphere. As ‘stable’ is a relative term, we computed the average number of types of bonds for molecules that have half-lives of 1, 10^3^, and 10^6^ s. The results are shown in Figure 6.

Each of the sets of molecules has a characteristic diversity of bond types, the ChEMBL set having the most diverse set of bonds per molecule, and the natural products set having the least diverse set of bonds per molecule. The apparently reduced diversity of bonds in Natural Product molecules is not because the Natural Products database contains smaller molecules than the other databases (Figure 6, right panel). The relative paucity of bond types in natural products is likely to be an effect of the near exclusion of some bond types from biochemistry, bond types that are widespread in synthetic chemicals (e.g., [86,95,96]).

We compare the number of types of bonds in molecules that are stable at 45 km on Venus (high temperature and high acid concentration) with the number of types of bonds in molecules that are stable at 65 km (low temperature, lower concentration). If hot, concentrated sulfuric acid caused instability of specific bonds, then we would expect the bond diversity at 45 km to be less than that at 65 km. Surprisingly, we did not find a significant change in the average number of types of bond per *stable* molecule with increasing CSA concentration and temperature as altitude is reduced. Note that this is the bond complexity of the molecules *that are stable* under those conditions. The *number* of molecules that have a threshold stability will be dramatically reduced, as shown in Figure 3. This suggests that the overall bond diversity remains the same at each altitude.

The results in Figure 6 suggest that there is significant structural diversity of chemistry possible in CSA and in consequence, CSA could, in principle, support sufficient theoretical chemical diversity to build biochemistry.

Figure 7 shows an analysis of the abundance of elements in compounds from the Sigma, ChEMBL, and PAN databases (i.e., not biochemicals), that are stable at different altitudes, weighted for the size of those databases. Biochemicals were not included because the elemental pallet used by life is more restricted than that used by industrial chemistry (e.g., using halogens rarely and not using silicon at all in small molecules), and so the Natural Product dataset has inherent biases in elemental use that make it incomparable to the other sets for this purpose. Figure 7 confirms that the molecules that are stable at different altitude have a similarly diverse set of elements, with two exceptions.

Compounds of phosphorus and silicon are substantially under-represented in the molecules that are stable in CSA compared to the data sets on which this analysis is performed. The instability of phosphorus compounds, which are overwhelmingly acid-hydrolysable esters in these datasets, is expected. Some aromatic phosphines are highly stable in sulfuric acid, but there are few representatives of these in the datasets used here. More unexpected is the depletion of silicon-containing compounds, as our previous analysis [87] suggests that a greater diversity of organosilicon compounds should be stable in sulfuric acid than in water. We hypothesize that the relative instability of silicon compounds seen in Figure 7 is the result of the use of specific, selected silicon chemistry in organic synthesis, and is not representative of the more broad qualitative analysis of the stability of organosilicon chemistry done in [87]. Specifically, the large majority of silicon-containing compounds in the chemical dataset are trimethylsilyl ethers or compounds containing phenylsilyl groups, both of which are known to be acid-labile [97,98]. In both cases, silicon groups are being used as acid-cleavable protecting groups, and hence have been selected to be unstable in sulfuric acid. We tested this hypothesis by generating a set of molecules containing C, N, O, and H, with or without Si atoms, using the combinatorial chemical structure generator Combimol-B [87]. Molecules up to eight non-H atoms were generated. These sets of computationally generated molecules were then analyzed for potential stability in sulfuric acid according to the methods described in Section 3 and in [70]. The results shown in Figure 8 confirm that a more representative set of silicon-containing structures is predicted to show good stability at above ~58 km. The stability of C, N, and O-containing compounds in this computationally-generated set of molecules largely reflects the stability of compounds in the Sigma, ChEMBL, and PAN sets of molecules analyzed in Figure 7. This also suggests that only silicon-containing compounds are biased in these molecule collections with respect to their stability in CSA.

### 4.4. Stability of Prebiotic Chemistry in CSA

If CSA is to be considered as a solvent for life, then there must be a plausible path for life to occur in CSA, either by originating in CSA or colonizing CSA from another solvent. If neither an origination nor a colonization scenario can be envisaged, then CSA is, in effect, an uninhabitable environment, even if the calculations above indicate otherwise. This is a chemical analogy of the concept of ‘habitable but uninhabited’ environments isolated in space, which have been postulated for Mars [99]. Considering how life can arrive in a CSA environment is therefore a key part of considering whether CSA is a potential solvent for life.

To address the plausibility of life originating in CSA, we modelled the stability of chemicals that might be inputs to the origin of life (OoL) in CSA. While the route to OoL on Earth is still fiercely debated [84], all models postulate a sequence of abiotic chemical reactions of relatively simple organic molecules present in the environment that subsequently build the components of what could become life. The simple precursor molecules can be synthesized in situ in the preferred location, or provided externally. While external provision of simple OoL precursor molecules is plausible, in situ chemistry using those molecules is not plausible in CSA.

We analyzed the relative CSA stability of simple abiotic OoL precursor molecules as represented by the low molecular weight organics extracted from the Murchison meteorite (Figure 9).

Below the altitude of 50 km, few of the Murchison water-soluble molecules are stable, but above 60 km, a substantial fraction of the Murchison molecules are predicted to have stabilities of more than a few weeks. These include molecules containing C, H, N, O, and S (no P-containing molecule has been isolated from Murchison).

However, there are only extremely limited possibilities for these compounds to undergo OoL relevant abiotic chemistry under CSA conditions. We illustrate the paucity of reaction paths with the Allchemy software described by Wolos et al. [89]. Allchemy uses literature-derived reaction types to build networks of reactions, starting from arbitrary molecules and ‘reacting’ them under a wide range of conditions, then taking the products and reacting them again, and so on. As a model, we took H_2_S, NH_3_, H_2_O, CH_4_, formaldehyde and hydrogen cyanide, six commonly cited prebiotic chemistry precursors, and one or two Murchison low molecular weight molecules as inputs to Allchemy. We modelled five ‘generations’ of reaction chemistry starting from these precursors, under either ‘neutral’ (mildly acidic, neutral, or mildly basic) conditions, or ‘acidic conditions’, in water or protic solvents. Figure 10 shows the ratio of molecules predicted to be formed under acid conditions to those predicted under neutral conditions, and illustrates the paucity of potential chemical reactions under acid conditions that could lead to critical OoL precursor molecules.

‘Neutral’ conditions predict the production of hundreds of products, ‘acid’ conditions gave no more than five, and often only formic acid and formamide. Such computational predictions are likely to over-state the potential for forming organics in CSA, as formic acid is decomposed in CSA with a half-life of ~100 s at 15 °C [100] (the hydrolysis of formamide in CSA has not been measured).

Our findings that the OoL is unlikely in CSA are further supported by Miller–Urey type experiments. If conducted in neutral or oxidized atmospheres these produce negligible organics. Cleaves et al. [101] showed that negligible synthesis of organics was a result of acid formation in the water phase of the experiment—if the water was neutralized then organics were formed much more readily. We conclude that under conditions where oxidized gases (SO_2_, CO_2_) and acid solutions are present, Miller–Urey type chemistry likely would yield no or few organic products that could act as precursors to OoL.

## 5. Discussion

CSA is believed to be present in the clouds of Venus, and may be a common material on exoplanets [21]. Given speculation on alternatives to water as a solvent basis for life, we wished to explore the potential for CSA as a solvent for life. This paper presents a first analysis of the potential of CSA as a solvent for complex chemistry. Because the ability of CSA to dissolve diverse chemicals is well known, in this paper, we focus on the stability of molecules in CSA.

We find that a diverse chemical space can be stable in CSA. The CSA-stable chemical space includes small molecules that are soluble and insoluble in CSA, and potential amphiphiles. The chemical space also includes polymers that have a range of stabilities in CSA and which include materials that are soluble and ones that are insoluble in CSA. In regards to these criteria, therefore, CSA is a plausible solvent for a biochemistry. Interestingly, and as noted in [87], there is a greater diversity of stable silicon compounds in CSA than in water. This work only explored some aspects of the ‘SPONCH’ chemistry, supplemented with some halogen and organosilicon chemistry. The finding that silicon adds significantly to the diversity of chemistry stable in CSA is one example of how the chemistry of substances in CSA differs significantly from the chemistry of the same substances in water. Another such example is the chemistry of phosphorus species. Phosphorus oxyacids act as *bases* in CSA, forming protonated forms quite unlike those that are stable in water [102,103]. Similarly phosphine is protonated in CSA; even though its pKa as measured in water [104] suggests that it is too weak a base to be protonated in an acid with a Hammett acidity of −1.5 [24].

The diverse chemistry of phosphorus and the unexpectedly stable chemistry of silicon suggests that other chemistries not explored here, that go beyond the ‘classical’ SPONCH terrestrial biochemistry, might provide yet more chemical diversity that is stable in CSA even if such chemistry is not stable in water.

### 5.1. Chemical Function in Sulfuric Acid

Our exploration of the structural diversity present in molecules that are stable in CSA does not address whether these molecules afford the functional diversity necessary for life [1]. As discussed above (Figure 3, Figure 4 and Figure 5), the large majority of terrestrial metabolites are not stable in concentrated sulfuric acid. We briefly address the issue of function, and use three examples to illustrate that a range of functional alternatives to terrestrial, water-based biochemistry can be envisaged for a potential biochemistry in concentrated sulfuric acid.

Benner et al. comment, “the C=O (carbonyl) group is central to all of contemporary Terran metabolism” [105], but almost all aldehyde and ketone compounds are unstable in sulfuric acid, as are their cyclic acetal tautomers. However, the vinyl group (CC=CH_2_) has analogous reactivity in concentrated sulfuric acid to the reactivity of the carbonyl group in water, and has been suggested as having the potential to provide a functional replacement for carbonyls [105].

Benner has suggested that a genetic material in water must have a repeating backbone charge to function [105,106,107]. As CSA is a highly polar solvent, a similar constraint might be needed on a genetic material in sulfuric acid. In terrestrial water-based life, the negative charge on the phosphates in DNA and RNA play the role of a repeating backbone charge—however, phosphate esters are very rapidly hydrolyzed in CSA. On the other hand, quaternary ammonium compounds stably dissolve in CSA (e.g., [108]), amines dissolve to form charged species, and iodinium compounds such as methylene blue are also stable in CSA solution [109]. Polymers containing these or other groups would have a repeating backbone charge, albeit a backbone with a positive charge rather than the negative charge found on nucleic acids in terrestrial biochemistry.

A fundamental requirement for life is a source of free energy [1]. Redox gradients may be available as energy sources in the clouds of Venus [110], but light to power photosynthesis is the most abundant energy source. We note that CSA is capable of supporting photochemical interconversion of other transition metal complexes (e.g., Re(VIII)/Re(VI) [111,112]). ‘Magic photoacids’ are compounds that can be converted to super acids (high-powered H donors) by light, a property they retain when dissolved in CSA [113]. Iron/sulfur redox chemistry may also be stable and functional in CSA [63].

While these are only a few examples, they suggest that, as it is in the case of the diverse chemical structures, diverse chemical function can also be supported in CSA-stable chemistry.

### 5.2. Habitability of Venus’s Clouds

We analyzed the stability of compounds under the conditions of Venus’s atmosphere as a specific example of an environment where CSA is present in liquid phase. We find that the extent of stable chemical space declines rapidly below ~60 km, and that maximum CSA-stable chemical space is seen at in Venus’s clouds at approximately 60–65 km. Temperatures in the 60–65 km region are −10–−20 °C, where CSA will be liquid and only moderately viscous. We suggest that, purely from a standpoint of chemical stability and diversity in CSA, the ‘CSA habitable zone’ for life based on concentrated sulfuric acid rather than on water is located at 60–65 km altitude in Venus’s clouds, and not in the lower clouds.

Any cloud-based life faces the problem of being rained out of the clouds. On Venus, the surface is uninhabitable, and rainout means extinction. Seager et al. [114] suggested a solution to this. The droplets in Venus’s clouds that contain microorganisms fall below the cloud layer and then evaporate, leaving ‘desiccated’ spores in the lower haze layer. Gravity waves in the atmosphere can then raise these back to the cloud layer where they act as cloud condensation nuclei.

The current work points to the chemical challenges of such a model. The stability of any organic chemical would decline substantially due to an increasing concentration of CSA and increasing temperature, as a droplet fell through the clouds to the haze layer altitude (<46 km altitude). Any biochemistry that operates at 60 km (whether using water or CSA as a solvent) would have to make very substantial adaptations during the life cycle to withstand the degradation of its components in an environment where the average half-time of reaction of all our dataset chemicals with CSA is in the order of milliseconds. Whether such adaptations are plausible is work for the future.

### 5.3. Overlap of CSA-Stable Chemistry with Terrestrial Biochemistry

The results presented here suggest that while sufficiently diverse chemistry for life’s biochemistry is possible in CSA, chemistries considered relevant to the Origin of Life (OoL) on Earth are unlikely to function in CSA. It is possible that very different abiotic chemistry to that proposed for the origin of life on Earth could occur in CSA, leading to life. However, if no path for life to originate in CSA can be found, then despite the analysis above, CSA is, in effect, uninhabitable, unless a migration path from another chemical environment is possible.

Current models of Venus’s evolution suggest that it had a clement surface with liquid water for at least a billion years, and possibly as late as 700 million years ago [115]. An origin of water-based life on early Venus is therefore plausible. The results presented above suggest that the gradual substitution of water solvent with CSA solvent is implausible. The large majority of biochemicals, both the chemical space of all chemicals made by life and the ‘core’ of metabolism on which all terrestrial life depends, are unstable in CSA on a timescale of milliseconds (Figure 4 and Figure 5). To adapt to use CSA as a solvent, terrestrial life would have to change over ^2^/_3_ of its entire metabolism. The magnitude of the required adjustment of the entire biochemistry to the new solvent would be an immense hurdle of change to overcome, even if such a solvent substitution was implemented gradually over hundreds of millions of years. Life is astonishingly adaptable, and has found ways of adapting to almost every environment on Earth where water can be made to be liquid—but it stretches credibility to suggest that it could evolve to do without DNA or RNA, when it is the DNA and RNA that are evolving.

### 5.4. Limitations and Next Steps

The work presented in this paper is a preliminary study to demonstrate that it is possible to go beyond qualitative statements on alternative solvents for life and move to quantitative discussions of a solvent’s plausibility. A solvent for life has to have suitable chemical stability and solubility properties. As sulfuric acid is notorious for its reactivity, we have focused here on the stability of chemicals. For other solvents, solubility may be a more limiting factor in their suitability, and so should be addressed quantitatively, as was done e.g., in [87]. In principle, a solubility model for CSA could be constructed, for example using the LFER methods of Abraham [116,117]. However, such methods would have to distinguish between solubility of the native molecule, solubility as stable protonated or sulfated species, and reaction of a molecule to form soluble products. This is work for the future. In this paper, we provide a quantitative discussion of chemical stability of organic molecules in CSA only, supplemented with qualitative discussion of solubility and functionality.

We present a reactivity model based solely on what has been experimentally measured (described in detail in [70]). There is very limited data on the reactivity of molecules that are unstable in CSA at laboratory ambient temperatures but which are potentially stable at < 0 °C. There are no data on the stability of selenium compounds (selenium is a minor but important element in terrestrial biochemistry), and very little on phosphorus compounds. While qualitative statements on the stability of many groups that are unstable in water but potentially stable in CSA are available, e.g., for acid halides, there is often no quantitative kinetic information. If the possibility of life in a CSA-dominated environment is to be taken forward, these lacunae need to be filled.

In common with much literature on speculations on biochemistry, we have analyzed molecules in isolation, i.e., as if they were pure solutes in pure CSA. In reality, biochemistry is likely to consist of many chemical species that can spontaneously react together in a complex mixture. Components of terrestrial biochemistry can react spontaneously (for example, the reaction of sugars and amines [118,119]), but do so at slow rates compared to the reactions of metabolism at physiological temperatures. The degree to which reactions between molecules in CSA further limits the chemical space available for life cannot sensibly be addressed without both a general model for chemical reactivity in CSA (which does not exist) and a model for the biochemistry. This is also work for the future.

## 6. Conclusions

We have explored the potential of concentrated sulfuric acid as a solvent for life. We provide a framework for quantitatively evaluating the potential of a solvent as a support for life and apply this framework to concentrated sulfuric acid. We modelled the stability of several sets of chemicals in concentrated sulfuric acid (CSA), using the clouds of Venus as a test case. We find that the between ^2^/_3_ and ^3^/_4_ of terrestrial biochemicals are unstable at any altitude, with half-lives of <1 s. However, 70–85% of a wider chemical space represented by chemicals made by humans have potential stabilities of >10^3^ s in CSA at low (sub-zero) temperatures, the conditions prevailing above ~60 km in Venus’s clouds. The stable molecules contain diverse elements and bonds, suggesting substantial structural and functional diversity can be present in chemicals that are stable in sulfuric acid. A qualitative overview of available literature suggests that many chemicals, but not all, will be soluble, that amphipaths required for the formation of membranes and nanostructures could exist in CSA, and that both soluble and insoluble polymers could be stable in CSA. To this extent, all the criteria for CSA to be a solvent for life are met. However, we find that abiotic chemical reactivity in CSA is likely to be extremely limited, and so conceiving of a path for life to originate in CSA is correspondingly difficult. Substitution of one solvent for another during the course of evolutionary adaptation (from water to a CSA solvent) is likewise not inconceivable but is likely to be very difficult. We conclude that life can exist in CSA, but further work needs to be done to explore how life could originate in CSA.

## Figures and Tables

**Figure 1 life-11-00400-f001:**
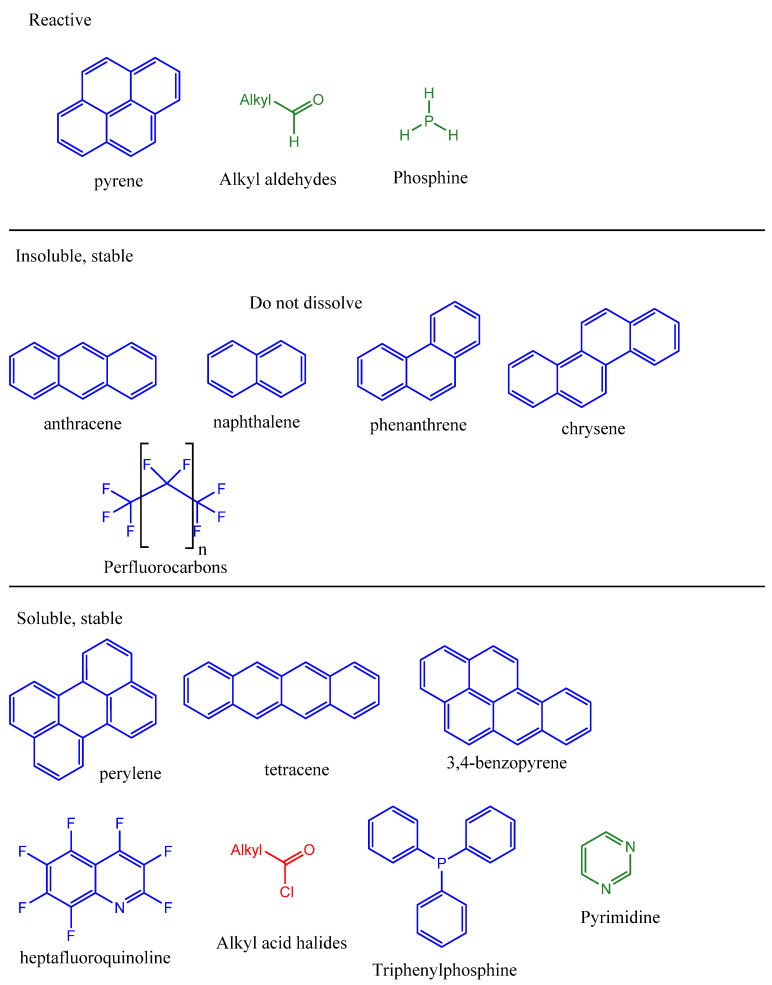
Examples of compounds and their fate in CSA; compounds that react rapidly (**top panel**), compounds that are insoluble (**middle panel**), and compounds that dissolve to give stable solutions (**bottom panel**). The panels refer to compounds fate in CSA, compounds are color-coded by their fate in water: red = liable to hydrolysis, green = stably soluble, blue = insoluble.

**Figure 2 life-11-00400-f002:**
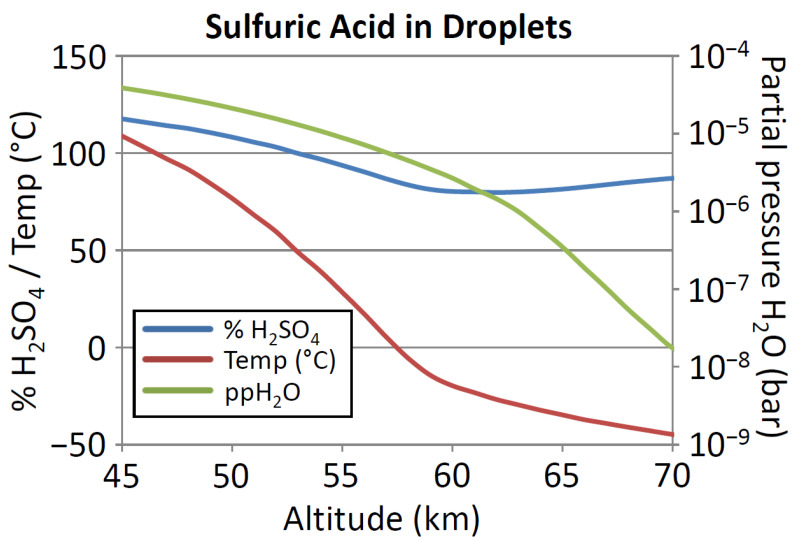
Concentration of sulfuric acid in cloud droplets in Venus’s atmosphere, and temperature and humidity. Calculations of acid concentration were performed as described in the text. Inputs in the calculation. *x* axis: altitude (km). *y* axis: left side % sulfuric acid (blue curve) and temperature (°C, red curve). Right side: partial pressure of water (pp H_2_O) in mmHg (green curve).

**Figure 3 life-11-00400-f003:**
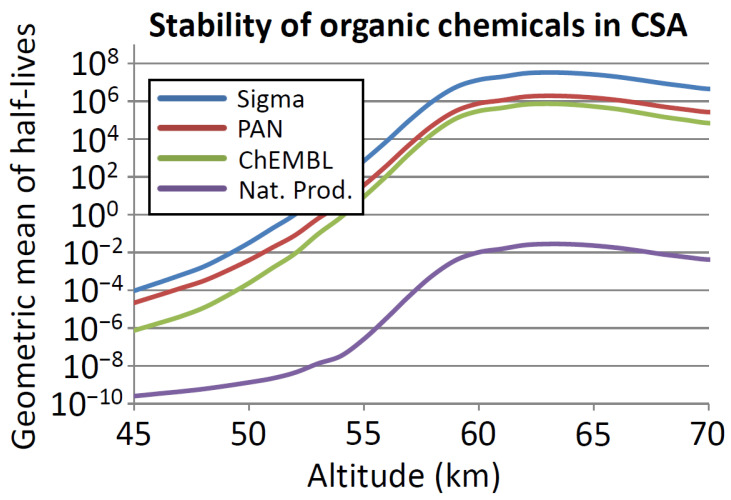
Average half-life of organic chemicals collected in several data repositories as a function of altitude. *y* axis: average half-life of chemicals in each of the four datasets as described in the Methods Section (Section 3.3). *x* axis: altitude (km). On average, half-life of biological molecules (Natural Products; purple curve) is much shorter in CSA than human synthetic compounds (Sigma; blue curve, PAN; red curve, ChEMBL; green curve). See main text for details.

**Figure 4 life-11-00400-f004:**
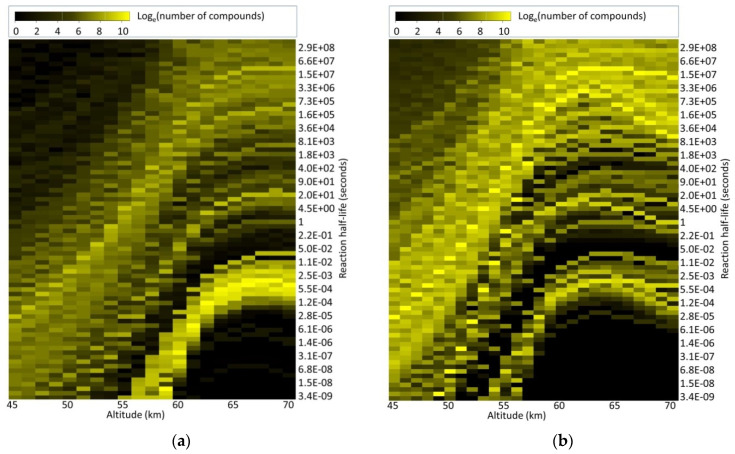
Fraction of chemicals that are stable as a function of altitude and half-life. *x* axis: altitude. *y* axis: the half-life of substances to reaction with sulfuric acid. Color axis: natural log of number of compounds in the chemicals database that has the *y* axis-specified half-life at the *x*-axis specified altitude. (**a**): prediction of half-life for Natural Products database. (**b**): prediction of half-life for ChEMBL compounds. On average, half-life of biological molecules (Natural Products) in CSA is much shorter than ChEMBL compounds (see main text for details).

**Figure 5 life-11-00400-f005:**
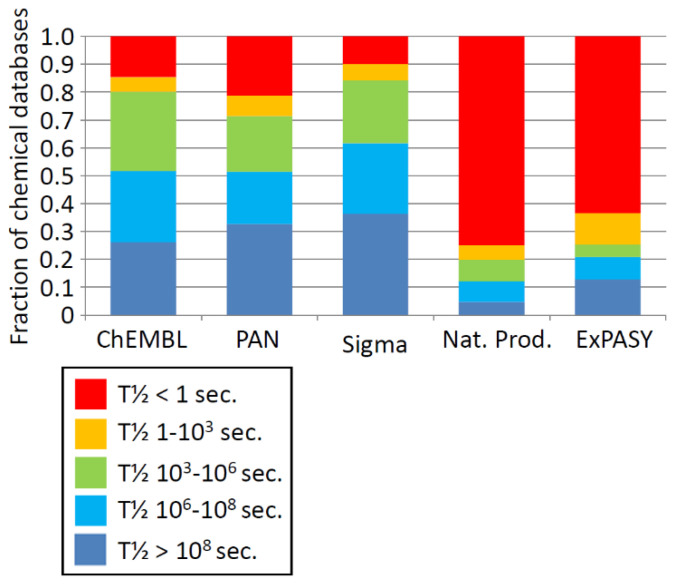
Fraction of chemical databases predicted to have threshold value half-lives. Averaged fraction of molecules that are ‘stable’ at different half-life (T½) thresholds as estimated above 62 km, for the four datasets, and the core metabolism ExPASY subset of the Natural Products dataset. Half-life of biological molecules (Natural Products, ExPASY) is much shorter in CSA than human synthetic compounds (Sigma, PAN, ChEMBL), and as a result much smaller fraction of biological databases is stable in CSA. See main text for details.

**Figure 6 life-11-00400-f006:**
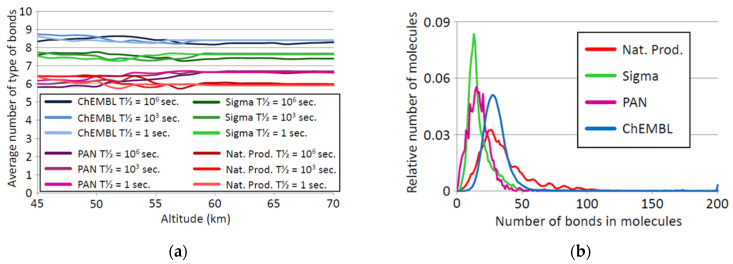
Bond diversity as a function of altitude on Venus. (**a**)*:* The average number of bond types in the molecules *that are stable* at each altitude was calculated for each of the four datasets, assuming that ‘stable’ meant a half-life of 1 s (“T = 1 sec.”), 1000 s (“T = 10^3^ sec.”) or 10^6^ s (“T = 10^6^ sec.”). *y* axis: average number of type of bonds in that set. *x* axis: altitude in km. Each set of molecules has a characteristic diversity of bonds within molecules, and that diversity of bonds is constant and the same in molecules that are stable at lower and at higher altitudes (i.e., at high concentration of CSA and high temperature and at low concentration of CSA and low temperature respectively). (**b**)*:* The distribution of the number of bonds per molecule in each of the four datasets. *y* axis: relative number of molecules in the dataset. *x* axis: number of bonds in the molecules *excluding* bonds to hydrogen atoms, so cyclohexane and benzene each have 6 countable bonds. Therefore, the bond number does not affect the stability profile of molecules in the investigated datasets.

**Figure 7 life-11-00400-f007:**
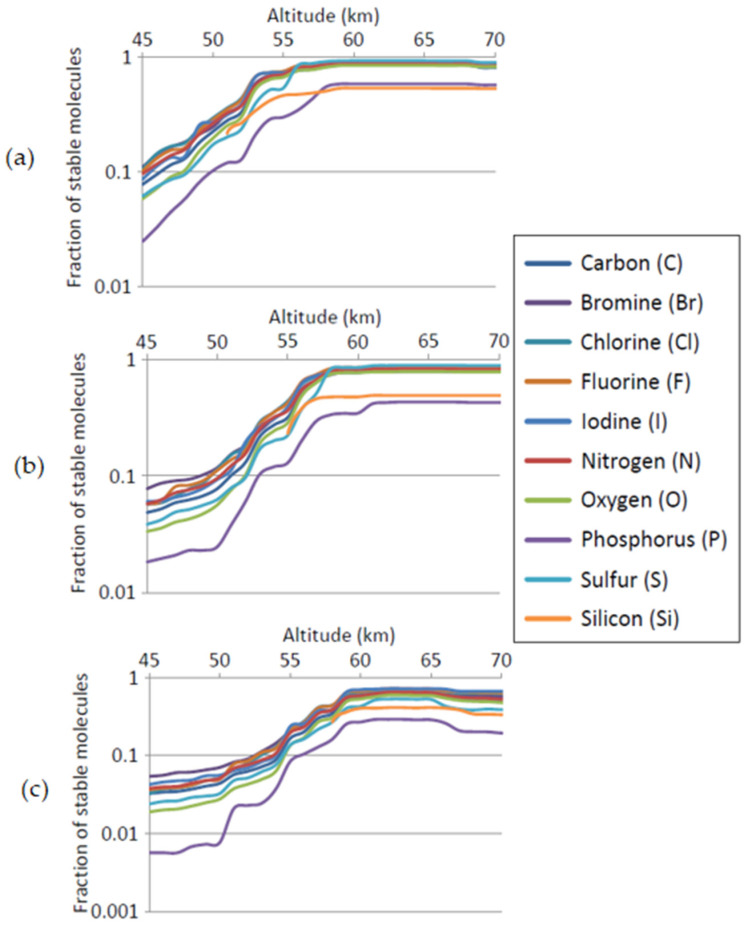
Distribution of compounds deemed to be stable by three stability criteria, by element present in those compounds. Compound set are average of Sigma, PAN, and ChEMBL datasets, weighted for size of the dataset. (**a**): stable = T½ > 1 s. (**b**): stable = T½ > 10^3^ s. (**c**): stable = T½ > 10^6^ s. In all panels: *y* axis = fraction of molecules in a dataset that contain an element (elements listed in the legend) that are stable at that altitude; *x* axis: altitude (km). Lines show values for different elements, as shown in the legend. The stability of chemicals in these datasets is not substantially related to their elemental composition *except* for the presence of silicon or phosphorus (as discussed in the text, see Figure 8 for further analysis of silicon).

**Figure 8 life-11-00400-f008:**
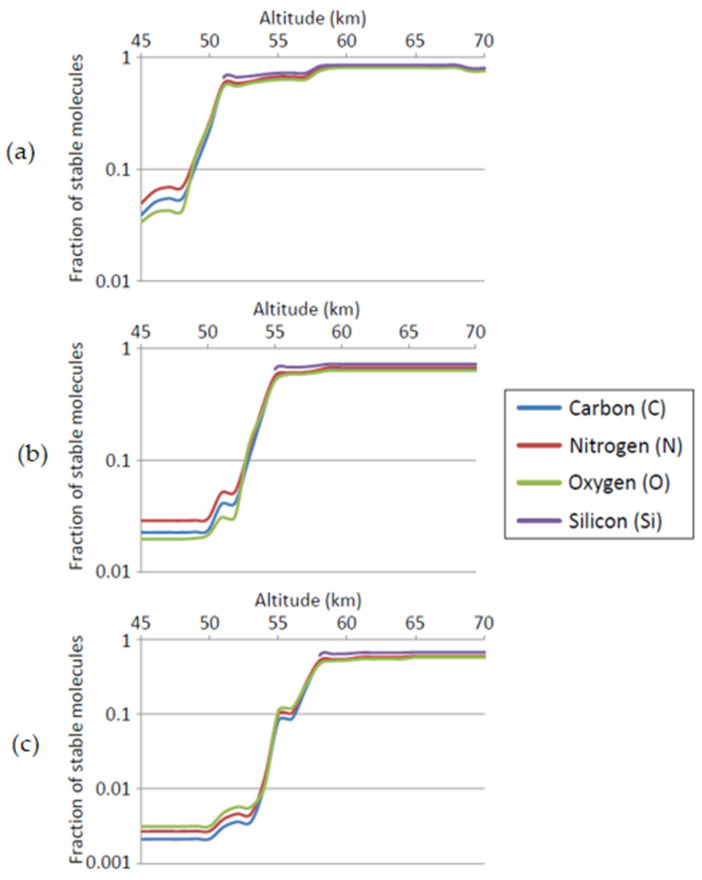
Distribution of compounds deemed to be stable (where stability is defined as having a half-life of at least 1, 10^3^, or 10^6^ s), by element present in those compounds. Data are 435,682 theoretical structures generated by Combimol-B [87] using elements C, N, O, H, and Si. (**a**): stable = T½ > 1 s. (**b**): stable = T½ > 10^3^ s. (**c**): stable = T½ > 10^6^ s. In all panels: *y* axis = fraction of molecules in a dataset that contain an element (elements listed in the legend) that are stable at that altitude; *x* axis: altitude (km). Lines show values for different elements, as shown in the legend. This shows that an unbiased sampling of the space of organosilicon molecules is no less stable in CSA than equivalent CHON chemical space.

**Figure 9 life-11-00400-f009:**
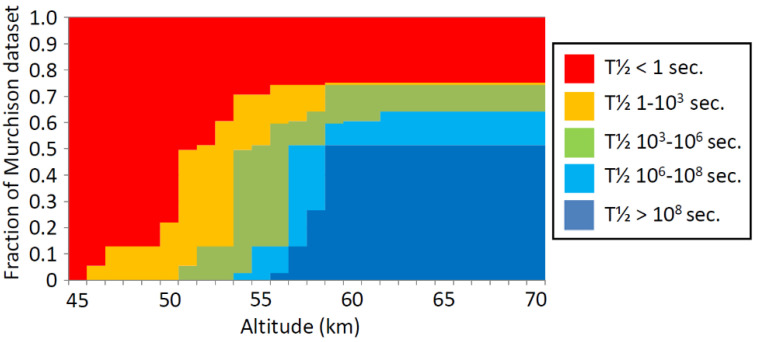
Stability of Murchison organics in CSA. The stability of soluble organic molecules from the Murchison meteorite as listed in [84] was modelled. *y* axis: fraction of the 109 molecules for which prediction could be made that are present in specific stability bands. *x* axis: altitude (km). Color represents the half-life of molecules in CSA at that altitude, with red meaning a half-life of <1 s, amber 1–10^3^ s, etc. Almost all Murchison small water-soluble molecules are unstable under Venus cloud base conditions, but a majority are stable at cloud top conditions.

**Figure 10 life-11-00400-f010:**
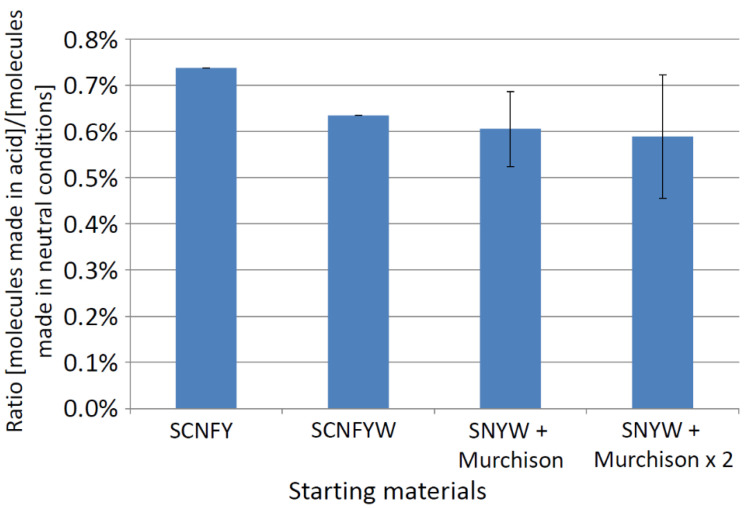
Comparison of acidic and neutral conditions for chemical synthesis. The Allchemy software was used to predict how many molecules could be made from specific starting materials in five steps or ‘generations’ of synthesis. *y* axis: ratio of number of molecules made under ‘acidic’ conditions to the number made under ‘mildly acidic’, ‘neutral’, or ‘mildly basic’ conditions. *x* axis: starting materials seeding the algorithm. C = methane, F = formaldehyde, N = ammonia, S = hydrogen sulfide, W = water, Y = hydrogen cyanide, Murchison = one molecule from the list of soluble organics found in Murchison (from [84]–117 molecules), Murchison x 2 = two molecules of soluble organics from Murchison, selected at random (1000 combinations). Error bars show one standard deviation on ratios (SCNFY and SCNSFW data points are a single value output).

**Table 1 life-11-00400-t001:** An overview of the databases where chemicals used in this study were drawn from. The Natural Products database was compiled from Dictionary of Natural Products (http://dnp.che4mnetbase.com (accessed on 18 July 2020)), UNPD (pkuxxj.pku.edu.cn/UNPD/introduction.php), Knapsack (http://kanaya.naist.jp/KNApSAcK (accessed on 18 July 2020)), and nine other databases each contributing less than 3% to the total [86]. The subset of natural products that are members of the core metabolism represented in the ExPASY metabolic map (https://web.expasy.org/pathways/ (accessed on 18 July 2020)) were also analyzed separately.

Database	Number of Compounds	Web Address	Ref.
ChEMBL	422,330	https://www.ebi.ac.uk/chembl/ (accessed on 18 July 2020)	[82]
Natural Products (NPs)	204,142	See Table legend for sources	[86]
SigmaAldrich catalogue	46,410	https://www.sigmaaldrich.com/ (accessed on 18 July 2020)	N/A
PAN	2866	https://pesticideinfo.org/ (accessed on 18 July 2020)	[83]

## Data Availability

Original chemical sets are available for download as Appendix A to this paper.

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
