# Peer review of "Evaluating Alternatives to Water as Solvents for Life: The Example of Sulfuric Acid"

_life, 2021, doi:10.3390/life11050400_

Round 1

Reviewer 1 Report

This is an interesting paper and analysis regarding the potential for concentrated sulfuric acid to serve as a solvent for life.  The narrative explicitly describes the chosen quantitative and qualitative analytical approaches as a “first pass” – though the analysis has the potential to provide a ‘first step’ – per se.  As an applied example, the narrative focuses on the altitude profile for sulfuric acid across Venus’ atmosphere, where abundances range from 70-98 % w/w (or perhaps 100% w/w as oleum), per the author’s descriptions. As a first step of addressing the potential for life in such high abundances, the manuscript focuses on the stability of biological and purely synthetic chemicals (I believe) in concentrated sulfuric acid.

The overarching coolness of this manuscript is the application of cheminformatics to theoretical astrobiology.  The application is seemingly novel, and the analytical approach is compelling.  

The strengths and weaknesses of the proposal are provided below along with associated suggestions.  A marked-up pdf of the manuscript has also been attached for reference regarding questions and minor issues.

The major strengths of the manuscript are the extracted structural themes for micro- and macromolecules that have appreciable lifetimes in concentrated sulfuric acid.  This component of the manuscript will be well-served with a deeper and more-focused chemical interpretation of the final results.  While the analytical approach seems sound, more information could certainly be squeezed or extracted for the informatics.

Major recommendations include (1) more explicit descriptions of the chemical databases and types of chemicals in the databases and (2) a more streamlined discussion and conclusion with an enhanced chemistry focus.

In the opinion of this reviewer, the major issues of the manuscript are (1) the reliance on halflife (zero order) to model reactions that are either first or second order (cation formation, sulfation, protonation, electrophilic aromatic substitution, etc.) and (2) the heavy emphasis put on the conclusions that life cannot transition into concentrated sulfuric acid, nor could evolve within such an environment, which opposes the premise of the manuscript – that concentrated sulfuric acid could be a solvent for life.

While the manuscript does a nice review of the impact of sulfuric acid in origins of life experiments, I don’t believe it’s necessary to conclude against sulfuric acid given the number of variables and regions of chemical space not yet experimentally considered.  While such consideration of origins and adaptations are critical components to astrobiology, the manuscript is perhaps overly burdened by such monumental questions.

Rather, a more streamlined focus is recommended to highlight the chemistry of this manuscript.

Major points:

  • For Figure 1, what is the measured parameter? Per my read of lines 363-364, water abundance was obtained from a reference, temperature from the VIRA model, and sulfuric acid abundance calculated from the provided equation. However, line 369 indicates the calculated concentration of sulfuric acid “gave” the partial pressure of water. Given the importance of this parameter, this section needs to be more explicitly described.
  • In Figure 2, is pressure included in the model, or just temperature?
  • Lines 562-572 seem like the crux of the manuscript; however, I was unable to follow the complete logic. Can this section be expanded and/or rewritten and/or rephrased?
  • Dialing back the conclusions that detract from the empirical nature of the manuscript.

Minor suggestions include:

  • Partial insolubility, rather than insolubility, leads to amphoteric properties.
  • Many polar chemicals used in biological applications have good solubility in DMSO. Certain enzymes are also soluble in DMSO.
  • Should the reduction of N2 under such conditions be considered exergonic, and not exothermic?
  • Do viscous glycerol solutions provide any insights into the analysis?
  • Line 242: should this be that kinetics cannot be determined without experimental insights?
  • Line 243-245, can this be broken up with an (A) and (B)?
  • Can line 245 be rephrased?
  • Are the 286 literature studies part of reference 48?
  • Can description of y-axes in Figure 3 legend be clarified?
  • Lines 725-726, is this referring to electrophilic aromatic substitution?
  • Figure 10 legend indicates that blue is insoluble, however, 4 of 7 soluble molecules are blue.
  • For solvation of metals, what role does redox-active sulfate play?
  • In Figure 11, what the temperature range for the gases?
  • Regarding redox gradients, a recent paper in GRL suggests a redox disequilibrium in Venus’ clouds.
  • Can lines 906-908 be rephrased?
  • Can lines 920-925 be rephrased?

Author Response

Response to reviewers.

We thank the reviewers for their comprehensive review of our paper. We provide specific answers to their points below. Their comments are in bold, our responses in plain text.  

As the Editor particularly asked us to look at Reviewer #3 we start with this review. Other reviews therefore refer to Reviewer #3’s comments in places.

Reviewer #1

This is an interesting paper and analysis regarding the potential for concentrated sulfuric acid to serve as a solvent for life.  The narrative explicitly describes the chosen quantitative and qualitative analytical approaches as a “first pass” – though the analysis has the potential to provide a ‘first step’ – per se.  As an applied example, the narrative focuses on the altitude profile for sulfuric acid across Venus’ atmosphere, where abundances range from 70-98 % w/w (or perhaps 100% w/w as oleum), per the author’s descriptions. As a first step of addressing the potential for life in such high abundances, the manuscript focuses on the stability of biological and purely synthetic chemicals (I believe) in concentrated sulfuric acid.

The overarching coolness of this manuscript is the application of cheminformatics to theoretical astrobiology.  The application is seemingly novel, and the analytical approach is compelling.  

We are grateful for these positive comments.

The strengths and weaknesses of the proposal are provided below along with associated suggestions.  A marked-up pdf of the manuscript has also been attached for reference regarding questions and minor issues.

We are grateful to the reviewer for taking the time to mark up the MSS. As far as we can see, there is no mechanism for us to download a file from the review section of the website. We hope that MDPI can provide it. Nevertheless we provide a point-by point response to the reviewer’s comments below.

The major strengths of the manuscript are the extracted structural themes for micro- and macromolecules that have appreciable lifetimes in concentrated sulfuric acid.  This component of the manuscript will be well-served with a deeper and more-focused chemical interpretation of the final results.  While the analytical approach seems sound, more information could certainly be squeezed or extracted for the informatics.

Major recommendations include (1) more explicit descriptions of the chemical databases and types of chemicals in the databases and (2) a more streamlined discussion and conclusion with an enhanced chemistry focus.

Thank you for this point. Reviewer #3 also asked for more detail on the database, and we have supplied this and expanded the description of the used chemical databases in Section 4.3 of the Materials and Methods (and Table 1). We have also restructured the paper to provide a more streamlined presentation of the background information, results and the discussion. Thus, the paper is now structured as follows:
Section 1. Introduction
Section 2. A framework for evaluation of potential solvents for life
Section 3. Background on CSA as a potential solvent for life – which is a new section that we added upon reviewer #3 request. Section 3 collects and summarizes scattered literature on the properties of CSA (giving the paper enhanced chemistry focus, as the reviewer suggested) and organizes it in the context of the potential of CSA as a solvent for life.
Section 4. Materials and Methods
Section 5. Results
Section 6. Discussion
Section 7. Conclusions

We hope that this new arrangement of text and contents of the paper is clearer for the reviewer.

In the opinion of this reviewer, the major issues of the manuscript are (1) the reliance on halflife (zero order) to model reactions that are either first or second order (cation formation, sulfation, protonation, electrophilic aromatic substitution, etc.) and (2) the heavy emphasis put on the conclusions that life cannot transition into concentrated sulfuric acid, nor could evolve within such an environment, which opposes the premise of the manuscript – that concentrated sulfuric acid could be a solvent for life.

Half-life is of course first order kinetics (rate dependent on the concentration of reactant), and we discuss briefly in the revised version why this is appropriate for the large majority of the reactions involved in solvolysis. Of course strictly they are second order or higher, but as sulfuric acid is assumed to be present in excess, its concentration is effectively constant, and so these are pseudo-first order reactions and under these conditions can be treated as such and characterised by a half-life. Sulfonation is also frequently first order in sulfonation target in excess CSA and in the absence of substantial product (as desulfonation then renders the kinetics more complex). For examples see

https://pubs.rsc.org/en/content/articlehtml/1985/p2/p29850000659

https://pubs.acs.org/doi/pdf/10.1021/ma00050a017

https://onlinelibrary.wiley.com/doi/abs/10.1002/recl.19620811108

https://pubs.acs.org/doi/pdf/10.1021/ma00100a006

https://www.sciencedirect.com/science/article/pii/S0009250919301915

https://onlinelibrary.wiley.com/doi/abs/10.1002/recl.19650840503

We have expanded on this in Section 4.1. of the Materials and Methods.

While the manuscript does a nice review of the impact of sulfuric acid in origins of life experiments, I don’t believe it’s necessary to conclude against sulfuric acid given the number of variables and regions of chemical space not yet experimentally considered.  While such consideration of origins and adaptations are critical components to astrobiology, the manuscript is perhaps overly burdened by such monumental questions.

We take this reviewer’s (and Review #3’s) point on OoL studies, and have softened our conclusions on this.

Rather, a more streamlined focus is recommended to highlight the chemistry of this manuscript.

We hope the revised order of sections of the paper meets this requirement. See above for the new revised outline of the paper.

Major points:

For Figure 1, what is the measured parameter? Per my read of lines 363-364, water abundance was obtained from a reference, temperature from the VIRA model, and sulfuric acid abundance calculated from the provided equation. However, line 369 indicates the calculated concentration of sulfuric acid “gave” the partial pressure of water. Given the importance of this parameter, this section needs to be more explicitly described.

Apologies, this was poorly worded. The reviewer is correct, the water concentration was taken as a given and the sulfuric acid concentration that was consistent with it calculated. We have moved this aspect of the method to the Materials and Methods section (Section 4.2.), and reworded.

In Figure 2, is pressure included in the model, or just temperature?

We only consider temperature. The effects of pressure change of an order of magnitude (a seen between the top and base of the Venusian clouds) is small, and depends on the volume of the transition complex(es), which is a level of mechanistic analysis beyond this modelling. For volatile reagents the partial pressure and the partition (Henry’s) constant of the reagent determines the concentration in the CSA phase, but for pseudo-first order reactions does not change the rate of reaction in that phase. We have added some to this effect in Section 4., Materials and Methods.

Lines 562-572 seem like the crux of the manuscript; however, I was unable to follow the complete logic. Can this section be expanded and/or rewritten and/or rephrased?

We have reworded the paragraph referred to substantially, and we hope the result is clearer.

Dialing back the conclusions that detract from the empirical nature of the manuscript.

We have trimmed the discussion section to focus more on the results of the modelling, and specifically (as per the request of Reviewer #3) removed additional discussion of literature results unless they are specifically relevant to our model results.

Minor suggestions include:

  • Partial insolubility, rather than insolubility, leads to amphoteric properties.
  • Our point was the existence of insoluble molecules and soluble molecules points to the potential for partially soluble, amphipathic molecules. We have clarified this.
  • Many polar chemicals used in biological applications have good solubility in DMSO. Certain enzymes are also soluble in DMSO.
  • Indeed they are. As are almost all drugs, many secondary metabolites that have very poor solubility in water. This is the point of mentioning DMSO – it is too good a solvent.
  • Should the reduction of N2 under such conditions be considered exergonic, and not exothermic?
  • Yes, the reviewer is correct. We have corrected this.
  • Do viscous glycerol solutions provide any insights into the analysis?
  • In terms of reactions in viscous solutions? They could do, but we think that a digression into enzyme kinetics in glycerol would be exactly the sort of drift from CSA chemistry that Reviewer #3 asked us to remove.
  • Line 242: should this be that kinetics cannot be determined without experimental insights?
  • We have reworded this.
  • Line 243-245, can this be broken up with an (A) and (B)?
  • We would rather keep this as text, but have reworded it to make it clearer
  • Can line 245 be rephrased?
  • As above
  • Are the 286 literature studies part of reference 48?
  • Yes – we have now stated this explicitly
  • Can description of y-axes in Figure 3 legend be clarified?
  • We have changed this, I hope the result is clearer.
  • Lines 725-726, is this referring to electrophilic aromatic substitution?
  • Yes, sorry, our mistake, electrophilic attack
  • Figure 10 legend indicates that blue is insoluble, however, 4 of 7 soluble molecules are blue.
  • Blue means insoluble in water, solubility in the panels is in CSA. We have tried to clarify this
  • For solvation of metals, what role does redox-active sulfate play?
  • Really interesting question! However, again in deference to Reviewer #3, we would rather not delve into  detailed review of solvation of metals in CSA here, but rather just note that solvation is facile and that redox chemistry can be performed.
  • In Figure 11, what the temperature range for the gases?
  • We have removed Figure 11. For the reviewer’s interest, though, this was Venus sub-cloud atmospheric temperatures.
  • Regarding redox gradients, a recent paper in GRL suggests a redox disequilibrium in Venus’ clouds.
  • This is an interesting new result, which we are happy to cite.
  • Can lines 906-908 be rephrased?
  • We have rephrased this
  • Can lines 920-925 be rephrased?
  • We have reworded this

Reviewer 2 Report

This paper presents an in depth analysis, as far as it is possible with available data, on the suitability of sulfuric acid as an alternative to water solvent for Life.  The journal „Life” is a very good venue for such an endeavor as with no page constrains the authors are free to make the analysis as exhaustive as the see fit. 
In my opinion they had done an excellent job and the paper is ready to be published in basically the present form. 

There are few small issues, which authors should correct or address before the final print form.

lines 366-370:
the authors do not explain what the symbols „LN” stand for in the equation

line 600: 
is: compounds should stable in sulfur 
should be: compounds should be stable in sulfur 

in Figure 10:
„Perfluorocarbons” are written in color font (consistent with color coding of the structures) for no clear reason. Since chemical structures are color coded in that figure, this fact might be baffling for the readers. 

line 897:
is: study) it si unlikely 
should be: study) it is unlikely 

line 978:
is: NADH etc.. The 
should be: NADH etc. The 

Author Response

Response to reviewers.

We thank the reviewers for their comprehensive review of our paper. We provide specific answers to their points below. Their comments are in bold, our responses in plain text.  

As the Editor particularly asked us to look at Reviewer #3 we start with this review. Other reviews therefore refer to Reviewer #3’s comments in places.

Reviewer #2

This paper presents an in depth analysis, as far as it is possible with available data, on the suitability of sulfuric acid as an alternative to water solvent for Life.  The journal „Life” is a very good venue for such an endeavor as with no page constrains the authors are free to make the analysis as exhaustive as the see fit. 
In my opinion they had done an excellent job and the paper is ready to be published in basically the present form. 

We are grateful to this reviewer for this opinion

There are few small issues, which authors should correct or address before the final print form.

lines 366-370:
the authors do not explain what the symbols „LN” stand for in the equation

We have replaced this with “ln”, the more accepted abbreviation for ‘natural log’

line 600: 
is: compounds should stable in sulfur 
should be: compounds should be stable in sulfur 

we have corrected this

in Figure 10:
„Perfluorocarbons” are written in color font (consistent with color coding of the structures) for no clear reason. Since chemical structures are color coded in that figure, this fact might be baffling for the readers. 

We have corrected the color on the label,

line 897:
is: study) it si unlikely 
should be: study) it is unlikely 

The new vesion does not include this typo.

line 978:
is: NADH etc.. The 
should be: NADH etc. The 

We have changed this

Reviewer 3 Report

The manuscript discussed various possibilities of the role of Concentrated sulfuric acid (CSA) in the origin of life (if any) and as a solvent to support a plausible life in the clouds of Venus. The concept of the paper was very interesting and I was very curious to go through it. However, the manuscript suffered from many serious flaws. I am going to mention some major points/concerns here:

  1. My far most concern is the “type” of the paper as an “article”. Although the paper was submitted as an article but it seemed to review other literature in the results section. But also seemed to show some modeling and calculations which meant it’s an original research paper but the way it was written looked like a review paper? In short, I am not sure if it’s a review paper or an original research paper?
  2. Lines 52-57, I do not agree. Please consider revising that the detection of PH3 can be directly linked to life in the Venus clouds. This is indeed controversial and should either be completely removed or revised. It’s highly likely that through some other mechanism and possibly some (abiotic) mechanism this gas is generated. Although, it indeed has attracted many astrobiologists/chemists to understand the chemistry here but the research in inconclusive. So careful wording should be used before making any conclusions here.
  3. Lines, 62-64, This is another a claim. Just merely based on its relevance and abundance H2SO4 can’t be considered as "solvent for life". There are so many criteria for this phenomenon. These lines should be revised. I highly disagree.
  4. Lines 134-136, I am not sure why DMSO and liquid CH4 are discussed here that these are not plausible solvents for life. The statement contradicts point no 3 if liq. CH4 is present abundantly, it could also be considered as a life relevant solvent based on its ubiquity (Both points 3 and 4 do not support each other).
  5. Lines 145-150 are completely irrelevant here. I do not understand their purpose.
  6. I found sub-sections 2.1 and 2.2 to be completely irrelevant with consideration of CSA as a solvent. These should only the necessary information relevant to CSA and its potential as a solvent and other unnecessary example (siliscate magma etc.) should be avoided or if possible both subsections should be deleted.
  7. As mentioned in point 6, the subsections 2.1 and 2.2 suffer from irrelevancy. These do not contain any necessary background information pertaining to the potential of H2SO4.
  8. Lines 230-231, 246-247, and 291-294 emphasize words such as "only an initial analysis", "we recognize this is an approximation" and finally lines 291-293 "this is a very rough approximation..." make the paper look very doubtful and weak and seem like the authors are not sure of their own work. Such instances should be revised.
  9. Lines 246-247, I did not understand the meaning of ,” We recognize that this is an approximation, but believe that it is fit for the purpose of this study”?
  10. Lines 308-310, were very confusing. These should be revised.
  11. Lines 309-312, the purpose of the reagents (cholinesterase, peptidases, proteases 310 or beta-lactamases) was not clearly discussed.
  12. A general observation while reading the manuscript, it was very confusing. The introduction, methods results and discussion were mixed with random data and discussion. The paper completely lacked focus and at times became very directionless. CSA, unfortunately was hardly the point of discussion.
  13. Chemical databases and software were not well explained, or references. The links to their websites were provided but no lists of the chemicals used in the current study were specified. The research’s aims and objectives were extremely confusing and extremely contradictory i.e. title suggested the potential use/plausibility of CSA as a solvent for life while the stabilities of a lot of irrelevant compounds were also given into consideration e.g. PAN Pesticide databases. Such studies should have been carried out separately.
  14. I would suggest the authors to delete all unnecessary subsections, choose their chemicals selectively e.g., origin of life/ biochemistry relevant and second category organics and other compounds other than relevant to life. The data sets were incomplete and hard to understand and did not have any enough strong conclusions in them.
  15. Lines 332-336 do not seem to correlate with the databases explained above. How the dissolution of various organic compounds in CSA would be related to the Venus clouds and other environments. The objectives of the research were very ambiguous.
  16. I did not quite understand and was rather surprised to see the inorganization of the manuscript. In the result section throughout the results of the authors were mingled with the previous work and irrelevant citations. The results section should only contain information of the presented work with only specific other citations e.g., 339-350, complete sub-sections 4.2 and 4.2.1. These need to be deleted or these should stay if the paper is a review paper?
  17. The equation/formulas discussed in the results section should perhaps move to method section?
  18. Line 355, Phosphorus is misspelled.
  19. Lines 372-373, please consider revising the grammar of the sentence.
  20. Once again sub sections 4.2 and 4.2.1 have irrelevant discussion in them. The results section should mainly contain the findings of the presented work. I am not sure the relevancy of these sub sections with the results.
  21. A very important question about results section. How would this answer the plausibility of CSA as potential solvent for the origin of life?
  22. It would be great if some chemicals were hand-picked and their stabilities and dissolutions were checked. The long list of databases is very ambiguous and confusing. Atleast a table should have been made to name all the major chemicals in the study.
  23. The key section of the whole research paper was section 4.5 that had Qualitative assessment of solubility of chemicals in concentrated sulfuric acid but unfortunately it was quite brief.
  24. Please consider totally deleting section 4.7, “Other properties of sulfuric acid” this did not seem to be part of the present research. It was irrelevant. Specifically, solvation of the metals is completely irrelevant with the chemistry of H2SO4 on Venus.
  25. Lines, 843-844, with ref to line 694 CSA is unlikely solvent for OoL and how it can be justified " as a plausible solvent" for the biochemistry while it could not be justified as a potential solvent for OoL? These two contradict each other.
  26. Lines 900-901, Although the solubility of chlorophyll in CSA is highly anticipated but speculations should be avoided
  27. Sub-section 5.2, “Habitability of Venus” is controversial and unconcluded discussion and mere discovery of PH3 in Venus atmosphere is NOT enough to justify the life. Hence the claims made in the section 5.2 (especially first paragraph) that are not concluded by data or supported by any experiments or calculations should be avoided no matter how strong these are felt to be true.
  28. I would suggest authors to completely rewrite the paper. Perhaps from scratch and remove all the unnecessary sections and focus on relevancy of CSA as a solvent for biochemistry/life/venus rather than unnecessary discussions. Clearly rewrite introduction, experimental, results (should mainly have only the current work’s results) and discussion.

Author Response

Response to reviewers.

We thank the reviewers for their comprehensive review of our paper. We provide specific answers to their points below. Their comments are in bold, our responses in plain text.  

As the Editor particularly asked us to look at Reviewer #3 we start with this review. Other reviews therefore refer to Reviewer #3’s comments in places.

Reviewer #3

The manuscript discussed various possibilities of the role of Concentrated sulfuric acid (CSA) in the origin of life (if any) and as a solvent to support a plausible life in the clouds of Venus. The concept of the paper was very interesting and I was very curious to go through it. However, the manuscript suffered from many serious flaws. I am going to mention some major points/concerns here:

  1. My far most concern is the “type” of the paper as an “article”. Although the paper was submitted as an article but it seemed to review other literature in the results section. But also seemed to show some modelling and calculations which meant it’s an original research paper but the way it was written looked like a review paper? In short, I am not sure if it’s a review paper or an original research paper?

We understand this reviewer’s concerns. A paper of this sort, the first to consider concentrated sulfuric acid as a solvent for life, must of necessity review a substantial amount of chemical literature to ‘set the scene’. The submitted version of the paper considered the properties of a solvent in major sections, and within each covered review and research results. In light of the reviewer’s comments we have substantially restructured the paper as described under Point 12 below.

We would regard this as a research paper. Because this is the first time the concept of sulfuric acid as a solvent for life has been explored, the literature background has to be explicitly laid out rather than referring to previous papers. This has been done in a new section (Section 3) as discussed below.

  1. Lines 52-57, I do not agree. Please consider revising that the detection of PH3 can be directly linked to life in the Venus clouds. This is indeed controversial and should either be completely removed or revised. It’s highly likely that through some other mechanism and possibly some (abiotic) mechanism this gas is generated. Although, it indeed has attracted many astrobiologists/chemists to understand the chemistry here but the research in inconclusive. So careful wording should be used before making any conclusions here.

A point of clarification is that we did not anywhere link PH3 to life in the Venus clouds. Our wording aimed to state the fact that the report of detection of phosphine in the clouds of Venus has revived interest in the possibility of life on Venus. One of us (WB) has carried out a study on sampling people’s opinion on Venus and this data and analysis in press, and supports our statement.  This was the point of referring to the report here. We have reworded this short segment to make this clearer.

  1. Lines, 62-64, This is another a claim. Just merely based on its relevance and abundance H2SO4 can’t be considered as "solvent for life". There are so many criteria for this phenomenon. These lines should be revised. I highly disagree.

We disagree. Ballesteros et al have suggested that liquid sulfuric acid may be common. Of course there are many criteria for a solvent for life other than the solvent’s possible existence. For this very reason we lay out some of those criteria in detail in section 2.  The point of mentioning it here is to argue that one of the criteria – that the liquid actually exists outside the lab, in the planetary environment – is met. We have added a clause to make this clear.

  1. Lines 134-136, I am not sure why DMSO and liquid CH4 are discussed here that these are not plausible solvents for life. The statement contradicts point no 3 if liq. CH4 is present abundantly, it could also be considered as a life relevant solvent based on its ubiquity (Both points 3 and 4 do not support each other).

Ballesteros et al estimate that liquid methane is even more abundant than liquid water in planetary systems, but, as we emphasise above, this is not a reason for considering it a valid solvent for water. Methane has been suggested as solvents for life in the past, and we have added citations to this. We have heard informally of the concept of DMSO as a solvent, but as we cannot find a citation for this we have deleted the comment. We also add a reference for the idea that magma or lava could be a solvent for life (mentioned below). As above, abundance is a reason for exploring a solvent as an alternative to water, not a reason for accepting it as a possible solvent for life.

  1. Lines 145-150 are completely irrelevant here. I do not understand their purpose.

The point was to discuss why modelling substances solubility   was not done in this paper. However we have removed the discussion of solubility modelling because the discussion, and review of solubility of molecules in concentrated sulfuric acid is now in a separate section (Section 3.2), and so there is no need to speculate on how modelling would be done, so we have removed this.

  1. I found sub-sections 2.1 and 2.2 to be completely irrelevant with consideration of CSA as a solvent. These should only the necessary information relevant to CSA and its potential as a solvent and other unnecessary example (silicate magma etc.) should be avoided or if possible both subsections should be deleted.

We strongly disagree with the reviewer’s comments on these two sections. The title of the paper is “Evaluating Alternatives to Water as Solvents for Life: the Example of Sulfuric Acid”. In order to evaluate anything, criteria against which it can be evaluated must be established. Section 2 establishes these criteria. If such criteria were commonly laid out in the literature on “Weird Life” then we would not have to lay them out here. However they are not. Rather, discussions of alternative solvents for life (even those that argue that there is no alternative, and water is the only reasonable solvent) focus on a specific property or a specific result. Thus for example, discussions of liquid methane/ethane cited above to not address solubility, they instead discuss how specific life-like chemistries such as amphipathic ‘membranes’ might occur in that liquid. We include citations to the use of methane and magma as solvents because these illustrate general principals. Specific examples illustrating a general point help comprehension of that point.

  1. As mentioned in point 6, the subsections 2.1 and 2.2 suffer from irrelevancy. These do not contain any necessary background information pertaining to the potential of H2SO4.

Again, we strongly disagree. Without a framework for evaluation, any paper is a set of ex cathedra statements, not a scientific argument. For well-established fields the criteria can be stated in a sentence or two and citations to the established literature. In the study of sulfuric acid as a possible solvent  we cannot make citations to general background, as there is no established literature available yet, so we have stated them explicitly.

  1. Lines 230-231, 246-247, and 291-294 emphasize words such as "only an initial analysis", "we recognize this is an approximation" and finally lines 291-293 "this is a very rough approximation..." make the paper look very doubtful and weak and seem like the authors are not sure of their own work. Such instances should be revised.

We are sure of our work, but do not want to over-state our case. The sections cited above relate to different aspects of the analysis, respectively calculation of chemical diversity, and two to the calculation of reaction rate. As this reviewer is well aware, the literature on ‘Weird Life’ is filled with absolute statements. Those statements may be justified by the data behind them, but in our observation they often are not. So here we wanted to be very clear what we are claiming. This is a first step.

Having said that, we agree that lines 291 – 294 (as in the original paper) might be over-cautious, and we have reworded these. We have also removed lines 226 – 232 (original numbering) as being a potential diversion, and replaced them with a more general discussion of what ‘chemical space’ means (see below in Point 13). 

  1. Lines 246-247, I did not understand the meaning of ,” We recognize that this is an approximation, but believe that it is fit for the purpose of this study”?

We apologise, and have reworded this.

  1. Lines 308-310, were very confusing. These should be revised.

    We address this point below in point 11.
  2. Lines 309-312, the purpose of the reagents (cholinesterase, peptidases, proteases 310 or beta-lactamases) was not clearly discussed.

We have reworded lines 308 - 312. To be clear, the specific enzyme assays used are not relevant. If a compound has been screened in a pharmaceutical assay, then it has been dissolved in water to be tested; this is a proxy for stability (and limited solubility) of molecules in water. We have tried to make this clearer in the text. See also below under Point 13.

  1. A general observation while reading the manuscript, it was very confusing. The introduction, methods results and discussion were mixed with random data and discussion. The paper completely lacked focus and at times became very directionless. CSA, unfortunately was hardly the point of discussion.

We are not sure what the reviewer means when they say “CSA, unfortunately was hardly the point of discussion.”  when “CSA” is mentioned 53 times in the Discussion section alone. However we take the broader point, and have tried to address it as follows. We have created a new section (Section 3) with literature review of specific properties of sulfuric acid. This new section is composed of sections and paragraphs moved from what was ‘Results’ and ‘Discussion’. Thus the paper is now structured as follows:
Section 1. Introduction
Section 2. A framework for evaluation of potential solvents for life
Section 3. Background on CSA as a potential solvent for life – which is a new section that we added upon reviewer’s request. Section 3 collects and summarizes scattered literature on the properties of CSA and organizes it in the context of the potential of CSA as a solvent for life.
Section 4. Materials and Methods
Section 5. Results
Section 6. Discussion
Section 7. Conclusions

We hope that this new arrangement of text and contents of the paper is clearer for the reviewer.

  1. Chemical databases and software were not well explained, or references. The links to their websites were provided but no lists of the chemicals used in the current study were specified. The research’s aims and objectives were extremely confusing and extremely contradictory i.e. title suggested the potential use/plausibility of CSA as a solvent for life while the stabilities of a lot of irrelevant compounds were also given into consideration e.g. PAN Pesticide databases. Such studies should have been carried out separately.

The description of the databases and the relevance of the databases are two separate topics, which we address separately here.

All the databases described are publicly available, and can be found with a simple Google search. All were referenced. However for convenience we have added a few clauses of description and a web links for each. The Natural Products database has been generated in-house at MIT from data that are proprietary and behind a ‘paywall’, so this cannot be linked; however we have also added a sentence description of this. Similarly, software was referenced, but we have added a short summary description here as well.
We have expanded the “Section 4.3. Chemical datasets and software” to include better description of the datasets used and the justification why chosen data sets are representative for the analysis that we present in the paper. We have also added Table 1. that provides an “at a glance” overview of the databases from which sets of chemicals used in this study were drawn.

The broader point is why chose these databases. We have added some text to explain our rationale in the Methods section.

  1. I would suggest the authors to delete all unnecessary subsections, choose their chemicals selectively e.g., origin of life/ biochemistry relevant and second category organics and other compounds other than relevant to life. The data sets were incomplete and hard to understand and did not have any enough strong conclusions in them.

As described, we did choose the example chemicals and the chemical data sets carefully. We are not trying to prove that terrestrial life can function in concentrated sulfuric acid, as it clearly cannot. Rather we are trying to explore the extent to which any chemistry that could be complex and chemically diverse enough to make up a biochemistry could be stable in sulfuric acid. The inclusion of databases of diverse chemicals that are not biochemicals (natural products – molecules produced by life on Earth) in this analysis is therefore a central part of the argument. For the same reason, a large set of diverse chemicals is required, not a small number of selected examples.

  1. Lines 332-336 do not seem to correlate with the databases explained above. How the dissolution of various organic compounds in CSA would be related to the Venus clouds and other environments. The objectives of the research were very ambiguous.

We have reworded the preamble to the Results section to try to make the goals of the presented research clearer.

  1. I did not quite understand and was rather surprised to see the inorganization of the manuscript. In the result section throughout the results of the authors were mingled with the previous work and irrelevant citations. The results section should only contain information of the presented work with only specific other citations e.g., 339-350, complete sub-sections 4.2 and 4.2.1. These need to be deleted or these should stay if the paper is a review paper?

As described above under Point 12 and Point 13, we have re-ordered the paper substantially. Our original concept was to group background literature and research results by the property – thus literature review and research results on stability were grouped together, those on solubility together and so on. However this reviewer felt very strongly that this was not a valid approach, and on reflection we understand their point. We hope the revised section order of the paper is satisfactory.

  1. The equation/formulas discussed in the results section should perhaps move to method section?

We agree and we have moved the equations and formulas to the Materials and Methods section (Section 4). We have also numbered the equations individually for ease of referencing.

  1. Line 355, Phosphorus is misspelled.

Apologies - corrected

  1. Lines 372-373, please consider revising the grammar of the sentence.

This is a split infinitive, which is considered to be poor form by some. We have changed this as requested. .

  1. Once again sub sections 4.2 and 4.2.1 have irrelevant discussion in them. The results section should mainly contain the findings of the presented work. I am not sure the relevancy of these sub sections with the results.

The ability to dissolve diverse chemicals is of course a critical feature of a solvent that could be considered as a solvent for a biochemistry, but we agree that this might not be the best place to discuss CSA’s ability to dissolve chemicals, and o we have moved this to the new literature discussion section (Section 3). See Point 12 for the new section outline of the paper.

  1. A very important question about results section. How would this answer the plausibility of CSA as potential solvent for the origin of life?

We rather hoped that our whole paper was addressing this question.. We hope the newly structured paper is clearer. Specifically, we show in our literature overview that the critical unknown in considering CSA as a solvent for life is the stability of a sufficiently diverse set of chemicals. The results section specifically addresses this, both for a broad chemicals pace and for the narrower question of chemcials believed to be relevant to the origin of life on Earth.

  1. It would be great if some chemicals were hand-picked and their stabilities and dissolutions were checked. The long list of databases is very ambiguous and confusing. Atleast a table should have been made to name all the major chemicals in the study.

It is not practical to list all the chemicals used in the study. There are about 800,000 chemicals collected from the publicly available databases and modelled in all (See the new Table 1 in the Materials and Methods section – subsection 4.3. for “at a glance” overview of the sets of chemicals used in this study).

Our paper is also a one of the first applications of big-data chemoinformatics to theoretical Astrobiology (a point also emphasized by another reviewer). Such an in depth analysis of the aggregated data is the focal point of any chemoinformatics approach. Addition of few hand-picked chemicals, while always useful as a targeted experimental validation of the stability model, adds only few “data points” to hundreds collected from the literature. Our work relies on over 250 literature reports of reaction kinetics, and as such we believe is a good start for such an analysis. We have published a separate paper on the collected data set of chemical reactivity of CSA with organic molecules “A Data Resource for Sulfuric Acid Reactivity of Organic Chemicals” that goes much more in depth into the approaches and methods used here. We agree that laboratory experiment would be valuable both to validate the models and to extend our knowledge of the solubility and stability of diverse chemcials in CSA. However that is a substantial experimental programme, and a completely different study from the present chemoinformatic approach. Our previous paper comments where such hand-picked laboratory work would be especially useful as a separate follow up study, where the literature data is incomplete.

  1. The key section of the whole research paper was section 4.5 that had Qualitative assessment of solubility of chemicals in concentrated sulfuric acid but unfortunately it was quite brief.

Solubility and stability are the two key properties. The reason that solubility is brief is that there is little question that some compounds are soluble in sulfuric acid, some are not, and some form amphipaths. The key question is what fraction of chemicals are stable in sulfuric acid. We have tried to make this clearer at the end of the new Section 3 (the literature review section) of the revised paper.

  1. Please consider totally deleting section 4.7, “Other properties of sulfuric acid” this did not seem to be part of the present research. It was irrelevant. Specifically, solvation of the metals is completely irrelevant with the chemistry of H2SO4 on Venus.

Solvation of metals may not be relevant to Venus’ bulk atmospheric chemistry (although there is evidence of volatile iron compounds in the atmosphere, and possibly dissolved FeCl2 in the cloud layer). The solvation of metals is critical to the chemistry of life. We have therefore left this section in because, as the title of the paper states, we are concerned with the chemistry of life in CSA, using Venus as a case study. We have however moved the “Other properties of CSA” section to the new Section 3 (as a subsection) for better flow and clarity of the paper.

  1. Lines, 843-844, with ref to line 694 CSA is unlikely solvent for OoL and how it can be justified " as a plausible solvent" for the biochemistry while it could not be justified as a potential solvent for OoL? These two contradict each other.

The two are a conundrum but not a contradiction. The current surface of the Earth is inimicable to almost all OoL scenarios, being powerfully oxidizing, but it is suitable for life. CSA appears to be incompatible with an OoL scenario like the ones envisaged for Earth, but as no work has been done on how life could originate in CSA, that is not surprising. We have tried to make this clearer in the revised paper. We have also clearly emphasized potential challenges of OoL scenario in the CSA,

  1. Lines 900-901, Although the solubility of chlorophyll in CSA is highly anticipated but speculations should be avoided

We do have evidence that chlorophyll is broken down by CSA from informal experiments done by one of the authors. We have amended this paragraph accordingly in the paper (the end of Section 6.1).

  1. Sub-section 5.2, “Habitability of Venus” is controversial and unconcluded discussion and mere discovery of PH3 in Venus atmosphere is NOT enough to justify the life. Hence the claims made in the section 5.2 (especially first paragraph) that are not concluded by data or supported by any experiments or calculations should be avoided no matter how strong these are felt to be true.

We are slightly confused by this. This section does not mention phosphine, and does not claim there is life on Venus. This section specifically addresses how our work on the stability of chemicals in CSA suggest where in the clouds it might be best to look for life that uses CSA as a solvent if such life were to exist. The reviewer might have confused Seager et al’s 2020 work on a potential solution to the ‘rainout problem’ with that group’s well-known involvement with the debate around the tentative detection of phosphine in the atmosphere of Venus, and what that might mean.  We have trimmed down this second part of the discussion, as we agree that it is a diversion from the main part of the paper.

  1. I would suggest authors to completely rewrite the paper. Perhaps from scratch and remove all the unnecessary sections and focus on relevancy of CSA as a solvent for biochemistry/life/venus rather than unnecessary discussions. Clearly rewrite introduction, experimental, results (should mainly have only the current work’s results) and discussion.

We thank the reviewer for their suggestion that we rewrite the paper. As suggested, we have restructured the paper completely (see Point 12 and Point 13), added some new material e.g. on the approaches and methods used (Section 4). We hope the result is more to the reviewer’s expectations.

Reviewer 4 Report

Interesting speculative preliminary contribution obviously driven by the Venusian phosphine. However, a medium for life is not just a passive solvent but also an active partner, driving the chemistry and organizing molecules and macromolecules. For example, the many virtues of water are often inventoried. The adaptation from water to CSA would be conceivable if CSA can behave as an effective reactant. Some comments on that aspect would be welcome.

Author Response

Response to reviewers.

We thank the reviewers for their comprehensive review of our paper. We provide specific answers to their points below. Their comments are in bold, our responses in plain text.  

As the Editor particularly asked us to look at Reviewer #3 we start with this review. Other reviews therefore refer to Reviewer #3’s comments in places.

Reviewer #4

Interesting speculative preliminary contribution obviously driven by the Venusian phosphine. However, a medium for life is not just a passive solvent but also an active partner, driving the chemistry and organizing molecules and macromolecules. For example, the many virtues of water are often inventoried. The adaptation from water to CSA would be conceivable if CSA can behave as an effective reactant. Some comments on that aspect would be welcome.

We agree with the reviewer that this is a very interesting question. We would like to speculate on this, but in light of the comments of Review #3, to which the Editor has particularly directed our attention, we feel that we cannot extend this aspect of the paper. Perhaps another paper could address how a water-based biochemistry could migrate to a sulfuric acid-base one. We have however included a paragraph in section 2.1 on how water acts as a reactant in terrestrial biochemistry, and the extent to which this can be stated to be a general requirement.

Reviewer 5 Report

The paper by Bains et al discusses the plausibility of sulfuric acid as a solvent for alternative life-forms (relevant to the atmosphere of Venus). The authors give a statistical prediction for the stability and solubility of a wide range of molecules and come to the conclusion that concentrated sulfuric acid indeed may host a sufficiently rich pool of molecules and chemical reactions necessary to form the foundations of life. On the other hand, by examining the stability and reactivity of molecules occurring in the Murchison meteorite, they come to the conclusion that sulfuric acid does not support the processes that could lead to the emergence of life. These are very interesting findings. In addition, the paper is well-written, therefore, in my opinion it definitely deserves publication in Life in its present form.   

Author Response

Response to reviewers.

We thank the reviewers for their comprehensive review of our paper. We provide specific answers to their points below. Their comments are in bold, our responses in plain text.  

As the Editor particularly asked us to look at Reviewer #3 we start with this review. Other reviews therefore refer to Reviewer #3’s comments in places.

Reviewer #5

The paper by Bains et al discusses the plausibility of sulfuric acid as a solvent for alternative life-forms (relevant to the atmosphere of Venus). The authors give a statistical prediction for the stability and solubility of a wide range of molecules and come to the conclusion that concentrated sulfuric acid indeed may host a sufficiently rich pool of molecules and chemical reactions necessary to form the foundations of life. On the other hand, by examining the stability and reactivity of molecules occurring in the Murchison meteorite, they come to the conclusion that sulfuric acid does not support the processes that could lead to the emergence of life. These are very interesting findings. In addition, the paper is well-written, therefore, in my opinion it definitely deserves publication in Life in its present form.   

We thank the reviewer for this very positive comment, and hope that the highly revised form still meets with their approval.

Round 2

Reviewer 3 Report

The manuscript by Bains et. al. has been revised well in some of the aspects such as the overall organization of the text, the sections, and sub-sections, the experimental sections as well as the result and discussion. I thank the authors for considering my suggestions. I also see that even rewording of the suggested text has been done as well e.g., relevancy of PH3 and the possible existence of life and a few other instances. This manuscript has gained much clarity. Also, I think the paper has some very interesting information that will be helpful for the origin of the life community. However, unfortunately, I still have some concerns and I am going to mention only the most significant ones:

  1. The abstract is incomplete and misleading. It suggests the potential study of SCA in the clouds of Venus. When the “biochemistry” word is used I am not sure which biochemistry? The life on Earth or the plausible Venus one? The abstract does not have a complete summary of the work e.g., no mention of studying the organics from meteoritic extracts. The abstract is indeed pretty straightforward, however, the text that follows is not. The abstract mentions the half-lives of the terrestrial biochemicals but does not mention the organics from the meteoritic extracts.
  2. Line 84, please correct the word “unknown”.
  3. As also suggested in points 6 and 7, I still think that sub-sections 2.1 and 2.2 suffer from irrelevancy with consideration of CSA as a solvent and the current paper. The actual paper and relevant contents in my opinion start from section 3 which is well written and relevant to the current topic. I would recommend deleting sections 2.1 and 2.2.
  4. Something minor, Lines 267 and 274 are repetitive. Please consider suggesting only once that CSA is a very strong acid.
  5. Lines 644 and onwards, I think this sentence is irrelevant here. I am not sure if some information cannot be provided then why it’s being mentioned here. I don’t think it would be of interest to the readers to know about something that they cannot have access to. This sentence should either be deleted or modified or perhaps described under the table as a special note rather than part of the main text.
  6. Table 1, I still have a hard time following the chemical database. The link leads to the website with an exhaustive number of chemicals. I am really wondering why would authors spend so much time on the general introduction and topics that were not even too relevant but here they will just provide a link that leads to the website and lead the readers to guess which chemicals. I would be really keen on seeing the results of rather a selective/limited number of chemicals rather than unlimited and unknown chemicals from the database (in bulk). How can a reader know which chemical? Indeed, the research is very interesting and unexplored previously. However, the choice of databases and the possible long list of chemicals is very hard to follow.
  7. I still recommend moving the irrelevant text from the results sections. For example, lines 690-701 suggest the presence of CSA in the clouds of Venus. This is not the result of the present work so either this should be moved to the introduction or otherwise in the discussion section. The results should strictly suggest the “results of the current/submitted work”.
  8. Something minor, line 726, “carry out” would probably be better used here as “carried”?
  9. Line 886, which authors the current authors? I suggest the use of the same word to self-phrase as "we", "us" rather than indirect speech? But this is again, minor. Also, in line 936, please change compare to past tense or otherwise keep it consistent.

  1. For the ease of understanding and more clarity, this paper should have either focused on 1) the Venus clouds (as abstract suggested) that would have included all the data sets suggested (including the stability of the meteoritic extract organics in CSA) or 2) alternatively focused on the idea of OoL and "prebiotic chemistry" with relevance to CSA. I think these sub-areas are themselves exhaustive and are worthy of separate publications.
  2. Lines 1032, I understand the exhaustive chemicals from the previously provided lists cannot be provided. This was one of the most important sections of the paper (stability of prebiotic chemistry in CSA) and it would be great if the list of the chemicals/organics extracted from the Murchison meteorite was provided which was easy to provide. I am not sure why this list was not given.
  3. Lines 1289 suggesting, “Our informal observations suggest that chlorophyll in CSA, we would expect chlorophyll to be unstable in CSA…is rapidly decomposed in CSA,….”

I do not suggest mentioning anything here without the results provided. If the authors have testified this, please provide the evidence, if not please consider deleting this.

  1. Lines 1321, “Venusian cloud-based life in which droplets that contain microorganisms…” Please consider rephrasing the sentence to “Venusian cloud-based life in which droplets that are hypothesized to contain microorganisms”. I do not think as of now we have any strong evidence besides PH3 discovery.

Author Response

The manuscript by Bains et. al. has been revised well in some of the aspects such as the overall organization of the text, the sections, and sub-sections, the experimental sections as well as the result and discussion. I thank the authors for considering my suggestions. I also see that even rewording of the suggested text has been done as well e.g., relevancy of PH3 and the possible existence of life and a few other instances. This manuscript has gained much clarity. Also, I think the paper has some very interesting information that will be helpful for the origin of the life community.

We thank the editor for his persistence.

However, unfortunately, I still have some concerns and I am going to mention only the most significant ones:

  1. The abstract is incomplete and misleading. It suggests the potential study of SCA in the clouds of Venus. When the “biochemistry” word is used I am not sure which biochemistry? The life on Earth or the plausible Venus one? The abstract does not have a complete summary of the work e.g., no mention of studying the organics from meteoritic extracts. The abstract is indeed pretty straightforward, however, the text that follows is not. The abstract mentions the half-lives of the terrestrial biochemicals but does not mention the organics from the meteoritic extracts.

We understand the issue. We have tried to keep in MDPI’s specified 200 word limit to the abstract, and so we have to summarise key conclusions and omit quite a lot of the argument and data behind those conclusions in the paper. We hope that we have made the abstract a clearer summary of the paper given this restriction,

  1. Line 84, please correct the word “unknown”.

    Corrected

  2. As also suggested in points 6 and 7, I still think that sub-sections 2.1 and 2.2 suffer from irrelevancy with consideration of CSA as a solvent and the current paper. The actual paper and relevant contents in my opinion start from section 3 which is well written and relevant to the current topic. I would recommend deleting sections 2.1 and 2.2.

    For reasons we have presented before, we strongly disagree with this view; these sections are relevant. If something is to be evaluated, then the criteria for its evaluation must be stated; to do otherwise is to reduce the paper to ex cathedra statements about analyses that have no root in argument. However as the editor is quite insistent on this, we have removed Section 2 as requested, replacing it with two sentences in the introduction. We have put the material (edited for context) in a Supplementary file, where it is available if the reader believes as we do that it is relevant, but where it does not disturb the flow of the main paper. We hope that this compromise is acceptable.

  3. Something minor, Lines 267 and 274 are repetitive. Please consider suggesting only once that CSA is a very strong acid.

We have tried to reword this.

  1. Lines 644 and onwards, I think this sentence is irrelevant here. I am not sure if some information cannot be provided then why it’s being mentioned here. I don’t think it would be of interest to the readers to know about something that they cannot have access to. This sentence should either be deleted or modified or perhaps described under the table as a special note rather than part of the main text.

    As requested, we have removed this and placed it in the Table footnote.

6. Table 1, I still have a hard time following the chemical database. The link leads to the website with an exhaustive number of chemicals. I am really wondering why would authors spend so much time on the general introduction and topics that were not even too relevant but here they will just provide a link that leads to the website and lead the readers to guess which chemicals. I would be really keen on seeing the results of rather a selective/limited number of chemicals rather than unlimited and unknown chemicals from the database (in bulk). How can a reader know which chemical? Indeed, the research is very interesting and unexplored previously. However, the choice of databases and the possible long list of chemicals is very hard to follow.

This is a chemoinformatics, “big data” approach, as one of the previous reviewers noted . Picking a few example molecules will be inherently misleading; you cannot derive a trend from a few examples, selected according to some arbitrary and hence biased criterion. Trends and statistical properties of a chemical space in chemoinformatics can only be derived from large numbers of molecules. So listing a few molecules from the collections would not be helpful to the reader and would not help others to replicate our results.

This is not an experimental paper, and so providing experimental results from a few chemicals would equally not be valid or helpful, and is not part of this work. However to help replication of this work and the editor’s (and future readers’)  understanding, we have provided all the datasets in SMILES format. SMILES is a universally used standard format in the field of chemoinformatics, and is more compact than other standards such as SDF. As noted before, we are not allowed by database contract terms to make the full NP dataset available. If the editor (or a reader) is not familiar with SMILES, the freely available Open Babel software can translate it into other formats.

  1. I still recommend moving the irrelevant text from the results sections. For example, lines 690-701 suggest the presence of CSA in the clouds of Venus. This is not the result of the present work so either this should be moved to the introduction or otherwise in the discussion section. The results should strictly suggest the “results of the current/submitted work”.

    The clouds of Venus are used as a specific example of an environment where it is plausibly believed that concentrated sulfuric acid exists. The literature provides a range of mutually inconsistent estimates of sulfuric acid concentration, which is why we calculate this profile which consistent with the water activity in the clouds. The calculation of the concentration of sulfuric acid in the clouds of Venus as a function of altitude (and hence of temperature) is a result of this paper. As this is a result, we believe that it belongs in the results section. To move it to the introduction would be to imply that this was a result derived from the prior literature, which is not true. We have therefore left this in the Results section.

  2. Something minor, line 726, “carry out” would probably be better used here as “carried”?

    Thank you, we agree and have changed this

  3. Line 886, which authors the current authors? I suggest the use of the same word to self-phrase as "we", "us" rather than indirect speech? But this is again, minor. Also, in line 936, please change compare to past tense or otherwise keep it consistent.

    We have changed this.

  4. For the ease of understanding and more clarity, this paper should have either focused on 1) the Venus clouds (as abstract suggested) that would have included all the data sets suggested (including the stability of the meteoritic extract organics in CSA) or 2) alternatively focused on the idea of OoL and "prebiotic chemistry" with relevance to CSA. I think these sub-areas are themselves exhaustive and are worthy of separate publications.

    We have added text to clarify why whether life can originate in clouds of sulfuric acid is an integral part of evaluating whether the clouds are habitable. If the clouds are theoretically habitable but there is no way for life to develop that can inhabit them, then they are in effect uninhabitable. For this reason OoL work is part of this study. Splitting the study into two would a) require a lot of duplication of text and explanation, and b) break the logical link between origin of life and existence of life. The editor will be familiar with arguments that the properties of water as a solvent for life reflect its unique role in life’s origin as much as in modern biochemistry. We need to point out that the same argument cannot (currently) be made for sulfuric acid.

  5. Lines 1032, I understand the exhaustive chemicals from the previously provided lists cannot be provided. This was one of the most important sections of the paper (stability of prebiotic chemistry in CSA) and it would be great if the list of the chemicals/organics extracted from the Murchison meteorite was provided which was easy to provide. I am not sure why this list was not given.

    We have provided the list of compounds isolated from Murchison. The original paper from which this list is derived (Bains 2020) provides original literature references for the detection of those chemicals in Murchison.

  6. Lines 1289 suggesting, “Our informal observations suggest that chlorophyll in CSA, we would expect chlorophyll to be unstable in CSA…is rapidly decomposed in CSA,….”
    I do not suggest mentioning anything here without the results provided. If the authors have testified this, please provide the evidence, if not please consider deleting this.

OK, we have removed it.

  1. Lines 1321, “Venusian cloud-based life in which droplets that contain microorganisms…” Please consider rephrasing the sentence to “Venusian cloud-based life in which droplets that are hypothesized to contain microorganisms”. I do not think as of now we have any strong evidence besides PH

    We have reworded this paragraph.